# Atomically precise engineering of spin–orbit polarons in a kagome magnetic Weyl semimetal

Hui Chen [1,2,3,7], Yuqing Xing[1,2,7], Hengxin Tan [4], Li Huang [1,2], Qi Zheng [1,2], Zihao Huang[1,2], Xianghe Han[1,2], Bin Hu [1,2], Yuhan Ye[1,2], Yan Li[1,2], Yao Xiao[1,2], Hechang Lei [5], Xianggang Qiu[1], Enke Liu [1], Haitao Yang [1,2,3], Ziqiang Wang [6], Binghai Yan [4] & Hong-Jun Gao [1,2,3] ✉

Atomically precise defect engineering is essential to manipulate the properties of emerging topological quantum materials for practical quantum applications. However, this remains challenging due to the obstacles in modifying the typically complex crystal lattice with atomic precision. Here, we report the atomically precise engineering of the vacancy-localized spin–orbit polarons in a kagome magnetic Weyl semimetal $Co_3Sn_2S_2$, using scanning tunneling microscope. We achieve the step-by-step repair of the selected vacancies, leading to the formation of artificial sulfur vacancies with elaborate geometry. We find that that the bound states localized around these vacancies undergo a symmetry dependent energy shift towards Fermi level with increasing vacancy size. As the vacancy size increases, the localized magnetic moments of spin–orbit polarons become tunable and eventually become itinerantly negative due to spin–orbit coupling in the kagome flat band. These findings provide a platform for engineering atomic quantum states in topological quantum materials at the atomic scale.

Creation and manipulation of many–body quantum states are crucial for developing advanced technologies in quantum computation, communications, security, and sensing[1-4]. Individual atomic-scale defects in a solid material provides one of the ideal candidates for generating localized quantum states due to the introduction of symmetry breaking, degeneracy lifting and scattering sources in the vicinity of the defects[5-11]. The atomically precise engineering of bound states has been realized in the vacancies of a few material platforms such as insulating film[12-14], diamond[15], graphene[16], h–BN[17] and two–dimensional transition-metal dichalcogenides[18], which is appealing for practical quantum applications. However, so far, the engineering of defect bound states is limited to a few material platforms due to the challenges in modifying the atomic defects of complex lattice.

Topological quantum materials have recently attracted considerable attention due to their fascinating symmetry-protected band structures and cooperative effects involving the interplay of multiple degrees of freedom (charge, spin, orbital, lattice)[19,20]. The interactions of multiple degrees of freedom in quantum materials are dynamically intertwined with each other, which results in exotic quantum states[21,22]. In recent years, the transition-metal kagome lattice materials which host Dirac points and nearly flat bands that naturally promote topological and correlation effects[23,24] are discovered, providing exciting

[1]Beijing National Center for Condensed Matter Physics and Institute of Physics, Chinese Academy of Sciences, Beijing 100190, China. [2]School of Physical Sciences, University of Chinese Academy of Sciences, Beijing 100190, China. [3]Songshan Lake Materials Laboratory, Dongguan, Guangdong 523808, China. [4]Department of Condensed Matter Physics, Weizmann Institute of Science, Rehovot 7610001, Israel. [5]Beijing Key Laboratory of Optoelectronic Functional Materials & Micro-Nano Devices, Department of Physics, Renmin University of China, Beijing 100872, PR China. [6]Department of Physics, Boston College, Chestnut Hill, MA 02467, USA. [7]These authors contributed equally: Hui Chen, Yuqing Xing. ✉e-mail: hjgao@iphy.ac.cn

opportunities for exploring frustrated, correlated, and topological quantum states of matter[25-33]. Remarkably, quantum states including the magnetic polarons have been discovered in magnetic transition-metal kagome shandites, which provides a promising way to engineer bound states for dilute magnetic topological materials and kagome-lattice-based devices[34,35]. On the other hand, the difficulty of reliability of engineering defects leaves it largely unexplored.

Herein, we report the atomically precise engineering of spin-orbit polarons (SOPs) localized at the S vacancies in a magnetic Weyl semimetal $Co_3Sn_2S_2$ by using low-temperature scanning tunneling microscopy (STM). The vacancies with well-organized geometry are precisely constructed though tip-assisted repairing method (Fig. 1a). Spin-polarized STM and magnetic field dependent measurements demonstrate the SOP nature of the bound states localized at the S vacancy. When the size increases, the bound states shift towards to the Fermi energy. In addition, the energy shift of bound states depends on the vacancy shape, which agree with the theoretical model based on the hoping between adjacent vacancies. Interestingly, as vacancy size increases, the magnetic states extend from localized magnetic moment to the negative magnetic moments resulting from spin-orbit coupling in the kagome flat band.

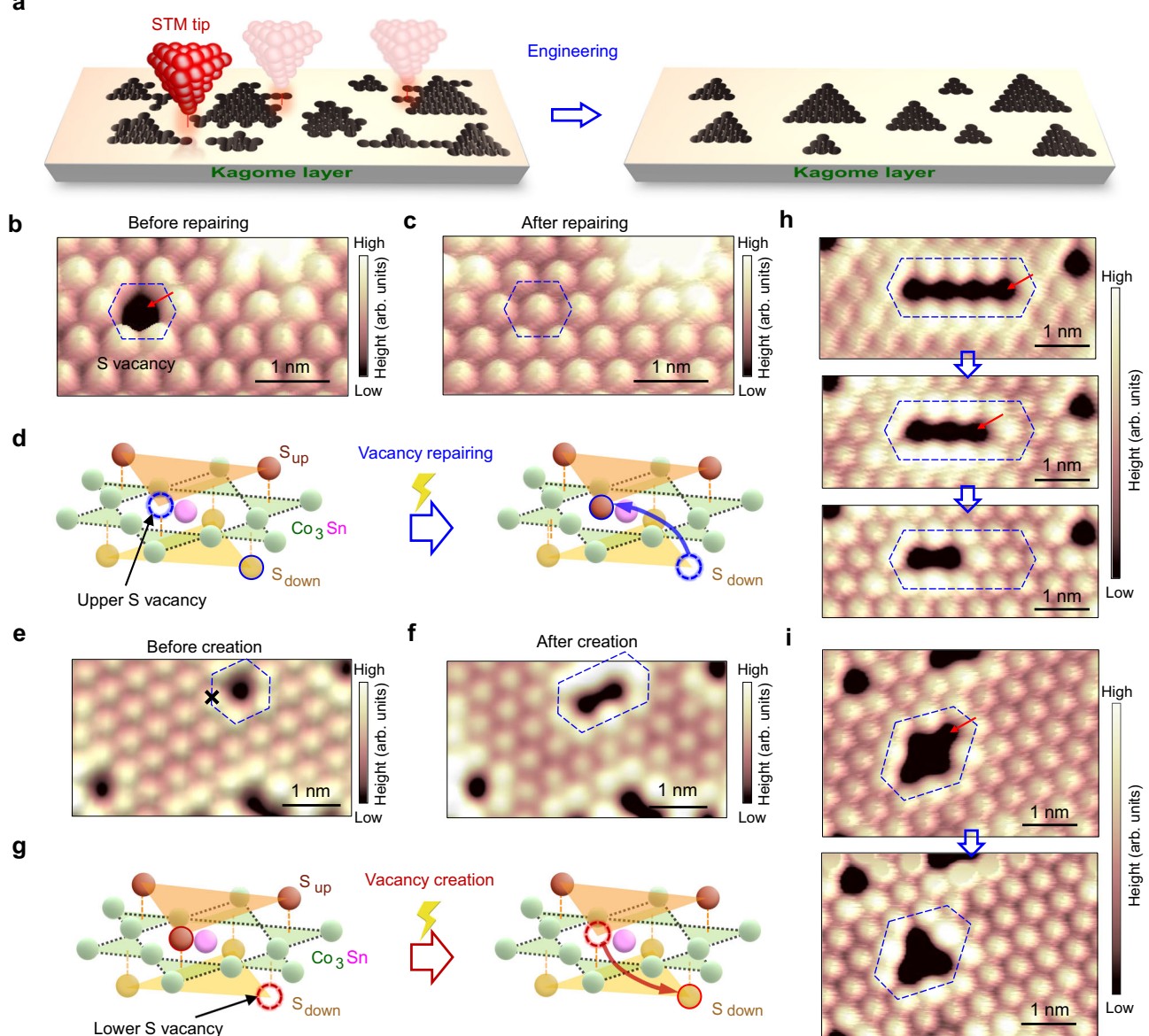

**Fig. 1 | Atomically precise engineering of vacancies at S-terminated surface of $Co_3Sn_2S_2$. a** Schematics of tip-assisted atomically precise vacancy engineering at a S surface over the kagome layer, showing that the vacancies with various shapes are transformed into the ones with well-organized geometries. **b,c** STM images showing the topography before (**b**) and after (**c**) vacancy repairing, demonstrating the filling of S atom. The red arrow indicates the position of tip pulse for the vacancy repair. **d** Schematic showing the filling of S atom from bottom S layer. **e, f** STM images showing the topography before (**e**) and after (**f**) vacancy creation, demonstrating the removal of S atom at surface. The black cross indicates the position of tip pulse for the vacancy creation. **g** Schematic showing the removal of S atom from top S layer to fill the vacancy at the bottom S layer. **h** Series of STM images showing that a long vacancy chain is gradually shortened by the vacancy repairing method. **i** Series of STM images showing that a cross-shaped vacancy consisting of four S absences leads to the formation of a triangular vacancy. The red arrows in (**b**), (**h**) and (**i**) mark the position of tip pulse during the vacancy repairing process. The black cross in (**e**) denotes the position of tip pulse during the vacancy creation process. STM scanning parameters for (**b, c**), (**e, f**) and (**h, i**): Sample bias $V_s = -400$ mV; Current setpoint $I_t = 500$ pA.

## Results

### Atomically precise construction of sulfur vacancies

The crystal structure of $Co_3Sn_3S_2$ consist of the rhombohedral lattice where the kagome $Co_3Sn$ layer are sandwiched between two triangular S layers, which are further encapsulated by two separated triangular Sn layers[32]. Cleavage in vacuum typically results in Sn and S terminated surfaces with kagome $Co_3Sn$ surfaces rarely obtained[29].

We start with the S terminated surface, which has been identified by STM and atomic force microscopy (AFM) in previous works[29,35,36]. The vacancies, which appear as hole-like features in STM images, are randomly distributed at S−terminated surface. The absence of single atom is further confirmed by the non-contact AFM (Supplementary Fig. 1). The vacancies consist of single vacancies and vacancy aggregates with various shapes (Supplementary Fig. 2).

We then achieve the atomically precise repairing of S vacancy through applying a voltage pulse from an STM tip. Figure 1b, c depict the experimental demonstration of repairing a single S vacancy. Briefly, to repair a single S vacancy, we position the STM tip close to the vacancy center (red arrow in Fig. 1b), followed by applying a tip pulse with a small voltage (position is marked by the red arrow in Fig. 1b, approximately ranges from 0.5 V to 1.0 V, independent of the pulse polarity). It is evident that the single S vacancy is filled with an additional S atom (highlighted by the blue hexagon in Fig. 1c). The filling of S atom is also confirmed by the non-contact AFM (Supplementary Fig. 3).

Prior to the vacancy repair, the d$I$/d$V$ spectrum obtained at a single vacancy exhibit a series of approximately equal-spaced spectral peaks, emerging just above the valence band inside the region of suppressed density of states. These peaks arise from a spin−orbit polaron localized around the single vacancy[35]. However, after applying the tip pulse, the d$I$/d$V$ spectrum obtained at the same position exhibits features identical to those of vacancy-free region near Fermi level, providing additional evidence that the S single vacancy is repaired by the tip pulse (Supplementary Fig. 4).

The origin of additional S atom to fill the single vacancy is illustrated in Fig. 1d. There are no topographic changes between the relatively large scale STM images before and after the tip pulse except for the vacancy repair (Supplementary Fig. 4). In addition, the diffusion energy for the migrating S atoms onto the S surface or STM tip is large due to the strong bonding between the surface S atoms and the underlying $Co_3Sn$ layer. The manipulation is achieved by using a clean tip which is immediately transferred into the STM chamber once calibrated at Au surfaces. Therefore, it suggests that the filling S atom originates from the underlying layers rather than the surface or STM tip. Considering that the crystal structure of $Co_3Sn_2S_2$ is composed of stacked…−Sn−[S−(Co₃−Sn)−S]−…layers, we define that top S layer of the sandwich structure corresponding to the as-cleaved S surface is the $S_{up}$ layer (brown triangle in Fig. 1d) while the underlying S layer of sandwich structure is $S_{down}$ layer (light brown triangle in Fig. 1d). Excited by the tip pulse, one S atom in the $S_{down}$ layer transfers to the vacancy site of the $S_{up}$ layer, resulting in the vacancy repair at S surface. In addition, the calculations indicate that the energy barrier for the S atom transfer from $S_{down}$ layer to $S_{up}$ layer is approximately 0.73 eV. The relatively low energy barrier means that it is possible to overcome it with a small voltage pulse applied at sufficiently small tip-sample distance (Supplementary Fig. 5).

There are two main features for the manipulation. (i) The vacancy repair at S surface is reversible (Fig. 1e, f, g). By putting the tip upon one of the upper S atoms (marked by the black cross in Fig. 1e) and applying a voltage pulse, we have achieved the vacancy creation at S surface (Fig. 1f). The success rate for the vacancy creation (about 5%) is much lower than repair (about 30%) in experiments. The lower success rate of vacancy creation further supports the proposed manipulation mechanism because the vacancy creation at top surface relies heavily on the existence of S vacancy around the creation position at the lower layer which is difficult to be identified by STM. (ii) Pulling up S atoms

from inner layer will create vacancies inside the crystal. These vacancies do not make observable contribution to the density of states at top surface states near Fermi level. The d$I$/d$V$ spectra between artificially-created and naturally-formed vacancies with the same shape on the top S layer show similar lineshapes near Fermi level (Supplementary Fig. 6).

The capabilities in controlled repairing of specific single vacancies provides a pathway for the controlled formation of vacancy in atomic precision. Motivated by this, we apply a step-by-step manipulation method to transform naturally-formed vacancy aggregates into artificial vacancy with well-defined shapes and sizes. For instance, we can manipulate the length of a one-dimensional vacancy chain by gradually filling S atoms into specific sites (Fig. 1h). Similarly, filling one S atom at a specific site of a cross-shaped vacancy consisting leads to the formation of a triangular vacancy (Fig. 1i). In more complex case, we are able to create quasi-regularly-shaped vacancy such as quasi-triangular and quasi-hexagonal vacancy by filling specific sites of vacancy aggregates with irregular polygon shapes (Supplementary Fig. 7).

### Coupling of bound states with controlled spacing of two neighboring S vacancies

The atomically precise construction of well-defined vacancy immediately provides an excellent opportunity to systematically investigate the evolution of the bound states with the vacancy size. We firstly study the simplest case of two spatially-separated single S vacancies with decreasing spacing (Fig. 2a and Supplementary Fig. 8). The d$I$/d$V$ spectra obtained around one single vacancy show that a series of approximately equal-spaced spectral peaks at −322, −300, and −283 mV gradually become suppressed as another vacancy approaches closer. Upon the formation of dimer vacancy, the series of bound states vanishes and a new series of sharp peaks at −292, −277, −265 and −254 mV appear (Fig. 2b).

To further study the evolution of density of states, we simultaneously collect the d$I$/d$V$ maps at the energy corresponding to bound states (Fig. 2a). Prior to the merging of two vacancies, the bound states in each vacancy exhibit localized flower-petal shaped patterns with three-fold rotation symmetry. After merging, the spatial distribution of the new four distinguishable d$I$/d$V$ peaks show two-fold rotation-symmetry patterns, with the shared S atoms connecting two single vacancies exhibiting the highest density of states (Fig. 2a and Supplementary Fig. 9). The peak located at−254 mV (Supplementary Fig. 9) is the sharpest and the most localized one, which is referred to the primary bound state.

The bound states of single vacancy have been demonstrated to emergent from the SOP[35], where vacancy-potential localized spin from the Co $d$ electrons and diamagnetic orbital current are trapped by the lattice distortion around the vacancies. Therefore, it is natural to investigate the bound states around the dimer vacancy. Thus, we further study the magnetic properties of the bound states around dimer vacancy through spin−polarized STM (details see Method). The d$I$/d$V$ spectra using a magnetic Ni tip demonstrate that both the primary bound states localized at single vacancy and dimer vacancy are magnetic with a spin-down majority (Fig. 2c). In addition, by applying a magnetic field perpendicular to the surface ($B_z$) and a non-magnetic STM tip, we observe anomalous Zeeman effect that primary bound states shift linearly toward the higher energy side independent of the field direction (Fig. 2d). We also study the lattice distortion around the S vacancies by applying a geometric phase analysis method[37,38] based on the Lawler−Fujita drift−correction algorithm[39] (Fig. 2e). The antisymmetric strain map $U(\boldsymbol{r})$ and symmetric strain map $S(\boldsymbol{r})$ show that the strain is mainly localized around the S vacancies, demonstrating the local atomic relaxations around vacancies. According to above evidences, we conclude that the bound states around dimer vacancy are originated from SOP as well[35].

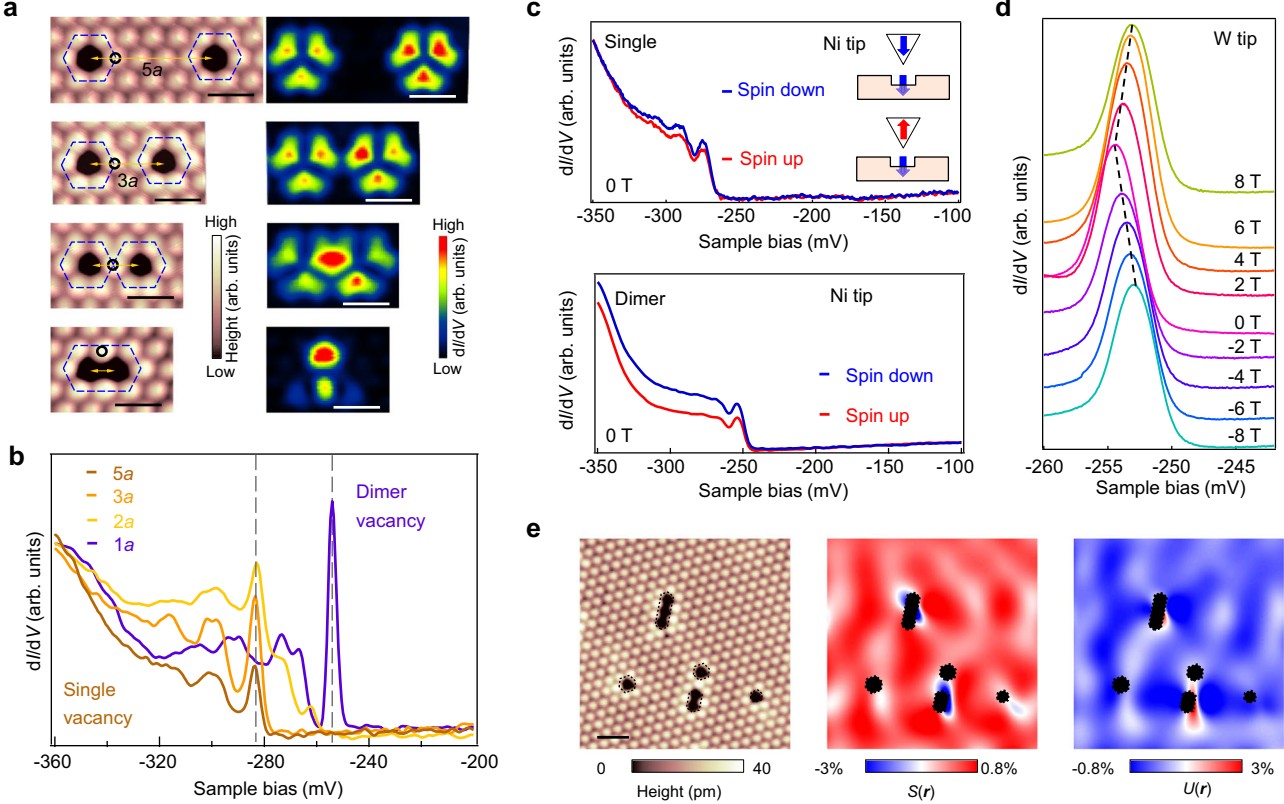

**Fig. 2 | Evolution of vacancy bound states with the spacings of two neighboring S vacancies. a**, Series of STM images, showing that two spatially-separated single S vacancies gradually merge into a dimer vacancy ($V_s = -400$ mV, $I_t = 500$ pA, $V_{mod} = 0.2$ mV). **b** The d$I$/d$V$ spectra obtained at a single vacancy of (**a**), showing the suppression of bound states as another vacancy approaches closer and emergence of new bound states for the dimer vacancy ($V_s = -360$ mV, $I_t = 500$ pA). The black dashed lines mark the energy position of bound states. The spatial positions for collecting d$I$/d$V$ spectra are marked by the black circles at each STM topographic image. **c** Spin–polarized d$I$/d$V$ spectra in the vicinity of single vacancy and dimer vacancy, showing a spin-down majority for both vacancies ($V_s = -350$ mV, $I_t = 500$ pA). Inset: schematics showing the magnetization of tip and the vacancy bound state. We define the spin direction as "up" and "down" with respect to cleaved S surfaces. **d** The d$I$/d$V$ spectra of the bound states in a magnetic field perpendicular to the sample surface from −8 T to 8 T, showing a linear shift toward higher energy independent of the magnetic field direction ($V_s = -300$ mV, $I_t = 2$ nA, $V_{mod} = 0.1$ mV). **e** Strain analysis of the S vacancies. The Symmetric strain maps $S(\boldsymbol{r}) \equiv u_{aa}(r) + u_{bb}(r)$ (middle panel) and antisymmetric strain map $U(\boldsymbol{r}) \equiv u_{aa}(r) - u_{bb}(r)$ (right panel) are derived from STM topography (left panel), showing that the lattice distorts around the vacancies. The vacancy regions are overlaid by black hexagons as there are no strains inside vacancies. The scale bars in (**a**) and (**e**) correspond to 1 nm.

## Size and shape dependent vacancy bound states

Inspired by the exotic bound states around dimer vacancy, we further study the evolution of the bound states with the length of vacancy chain. We construct a series of linear vacancy chain with atomic length $N$ using tip–assisted manipulation (Fig. 3a) and collect the d$I$/d$V$ spectra on the vacancies ($N = 1, 2, 3 ...$) respectively. All linear vacancies exhibit series of several peaks with equally spacing energy that emergent from the bound states. To facilitate comparison, we defined the sharpest one of the peaks with highest energy position to be the primary bound states of each vacancy ($P(N)$). We find that the $P(N)$ shift to the Fermi energy with increasing $N$, and eventually reaches a critical energy position at about -240 meV at $N > 4$, as highlighted by the black arrows (Fig. 3b and Supplementary Fig. 10).

In addition to the single-chain vacancy, the tunability of the bound states extends to the vacancies with more elaborate shapes, including double column vacancy chains (Fig. 3c), equilateral triangle (Fig. 3e and Supplementary Fig. 11) and equilateral-hexagon vacancies (Fig. 3g). All vacancies exhibit series of several peaks with equally spacing energy and the sharpest peak with highest energy position is similarly assigned as the primary bound states $P(N)$. As summarized in Fig. 3i, the evolution of $P(N)$ for each symmetric shape follows an exponential function, with all $P(N)$ shifting exponentially towards a critical energy value near the Fermi level as the size increases. The critical energy level of $P(N)$ depends on the vacancy shape (highlighted

by different color in Fig. 3), with higher symmetry shapes possessing higher critical energy levels (Fig. 3i). For instance, the critical energy of $P(N)$ of single chain vacancies is about −240 meV while one of the hexagonal vacancies is almost at Fermi levels.

The geometry dependent bound states suggest the strong couplings between adjacent single vacancies. Furthermore, the spatial distributions of the bound states across the single chain vacancies (Supplementary Fig. 12) exhibit quasi one-dimensional band behaviors[40–42], indicating the existences of vacancy–vacancy couplings. The d$I$/d$V$ maps at large-size triangular vacancy show quantum confinement effect, featuring a quantum antidot (Supplementary Fig. 13). To gain insight into the shape dependent energy shift behaviors of $P(N)$, we develop a simple model (see method in Supplementary Information and Supplementary Fig. 14) to simulate the bound states around vacancies. We simulate vacancies with a simple tight-binding model with a nearest neighbor hopping $t$ by considering the hybridization between vacancies (Fig. 3j). We construct four types of vacancy patterns with different number of S vacancies, which consist of single chain, double column chain, triangle and hexagon. In each vacancy configuration, the highest energy level is extracted as the vacancy state. We find that the vacancy state undergoes a similar exponential shift towards higher energy (Fig. 3k), which is consistent with experimental observations in Fig. 3i.

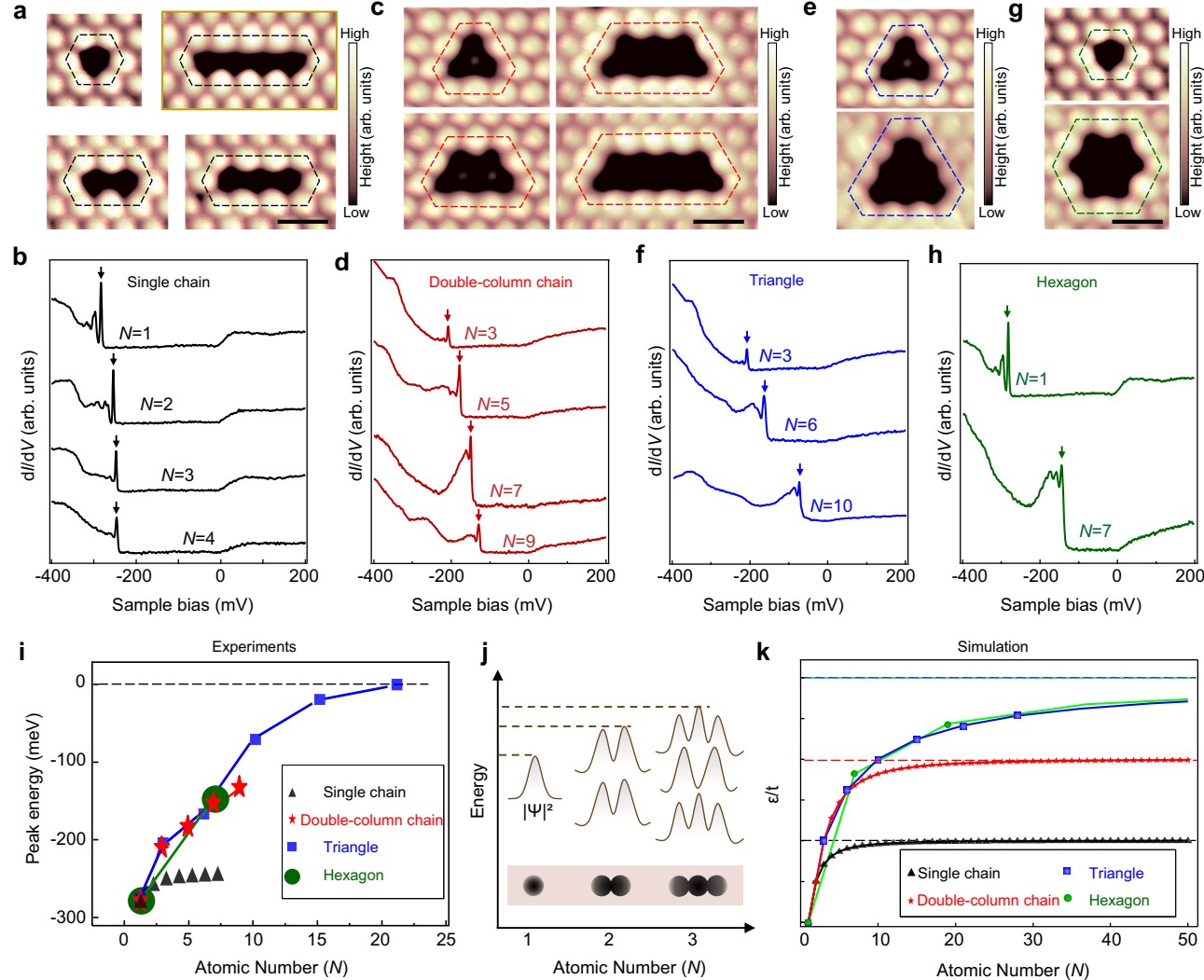

**Fig. 3 | Size and shape dependence of vacancy bound states. a–h** STM images and the d$I$/d$V$ spectra for linear vacancy chains (**a, b**), double column vacancy chains (**c, d**), triangular vacancies (**e, f**) and hexagonal vacancies (**g, h**), respectively. The peaks with highest energy position assigned as primary bound state for each vacancy are highlighted by the arrows in (**b**), (**d**), (**f**) and (**h**), respectively. **i** Evolution of primary bound states with vacancy size for different vacancy shapes, showing an exponential energy shift depending on the shape symmetry. **j** Schematic illustration of hybridization between vacancy bound states. Single vacancy creates bound states inside the gap-like density of states at S surface. The hybridization of the additional vacancy induces a new bound state at higher energy. **k** Calculated evolution of the bound state with the atomic numbers based on a tight-binding model with a nearest neighbor hopping $t$, showing similar energy shift with experiments in (**i**). The $\varepsilon$ is calculated energy levels of bound states based on the simple model. It indicates quantum confinement of the vacancy bound states. STM parameters for (**a–h**): $V_s = -400$ mV, $I_t = 500$ pA, $V_{mod} = 0.2$ mV. The error bars in (**i**) from multiple measurements on the same-geometry vacancies are smaller than 3 meV. The scale bars for (**a, c, e, g**) correspond to 1 nm.

## The size dependent magnetic moment of vacancies

To gain deep understanding of nature of localized states at vacancies, we further study the shape dependent magnetic moment of the bound states. We focus on the magnetic moment of triangular vacancy due to their high yields and relatively-large lattice distortions (Supplementary Fig. 15). The bound states of triangular vacancy present the anomalous Zeeman effect with external magnetic field (Fig. 4a, b). Fitting the energy position (details see Supplementary Fig. 16) as a function of the magnetic field, we obtained the effective magnetic moment value $|\mu(N)|$. For example, $|\mu(N = 3)| = 0.09$ meV/T $= 1.55$ μB (Fig. 4a) and $|\mu(N = 10)| = 0.17$ meV/T $= 2.93$ μB (Fig. 4b). These results indicate that the magnetic moment of bound states localized at vacancy is directly related to the vacancy size.

The Co₃Sn terrace, confined by the step edges of adjacent S terraces (Fig. 4c), is considered as a naturally occurring vacancy with an enormous size ($N = \infty$). The spatially-averaged d$I$/d$V$ spectrum obtained at the Co₃Sn terrace shows sharp peaks in the vicinity of

Fermi level, which is consistent with previous STM results on the Co₃Sn surface[29,36]. The magnetic field dependent d$I$/d$V$ curves (Fig. 4d) show similar anomalous Zeeman effect with an effective magnetic moment of $\mu(N = \infty) = -0.19$ meV/T $= -3.28$ μB (the negative value is aiming to highlight the moment is negative in anomalous Zeeman effect and differentiate it from the positive moment in Zeeman effect). The negative orbital magnetic moment results from the spin–orbit coupling in the kagome flat band considering the non-trivial Berry phase of the flat band[43]. Evolution of the magnetic moment with the atomic number of vacancies shows that the magnetic moments extend from localized magnetic moment around vacancies to the flat band negative magnetism from Co₃Sn kagome layer (Fig. 4e).

## Discussion

The atomically precise engineering of sulfur vacancies is not limited to ferromagnetic Co₃Sn₂S₂ but can extend to other Co₃Sn₂S₂ derived kagome metals. We have also achieved the controlled vacancy repair

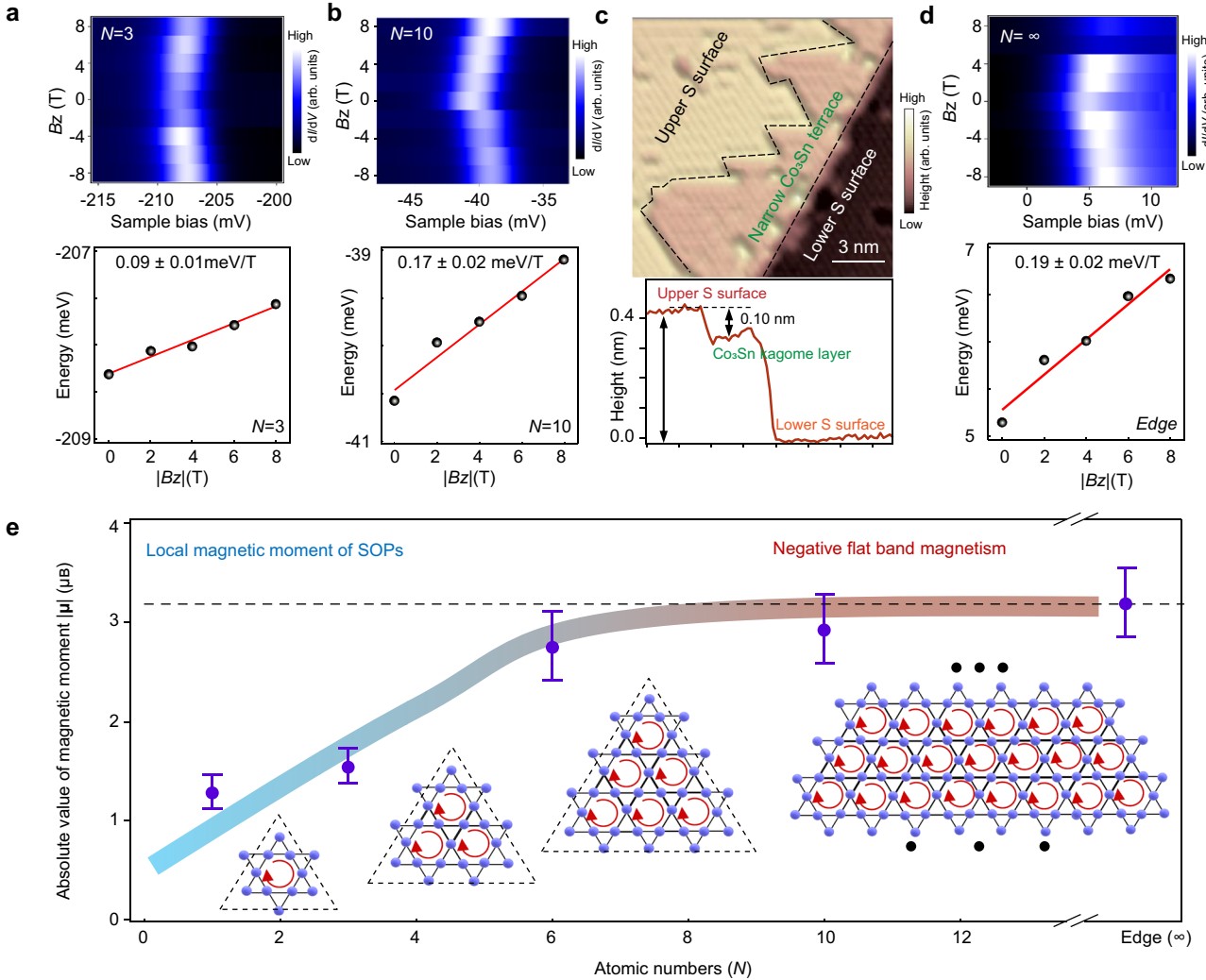

**Fig. 4 | Tunability of the magnetic moment of bound states through controlling the size of triangular S vacancies. a**, **b** Intensity plot of field dependent d$I$/d$V$ spectra (top) and corresponding energy shift of primary peak position with magnetic field (bottom) for triangular vacancy of $N = 3$ (**a**) and $N = 10$ (**b**). A linear function is applied to fit the energy shift of peak position, with the slope marked. **c** STM image (top) and corresponding line profile (bottom), showing a narrow Co$_3$Sn terrace confined by the step edges of adjacent S terraces. **d** Intensity plot of field dependent d$I$/d$V$ spectra (top) and corresponding energy shift of peak position with magnetic field (bottom) of the bound states at narrow Co$_3$Sn terrace in (**c**), revealing a magnetic moment of approximately 0.19 meV/T = 3.28 $\mu_B$ for the bound state near Fermi level. **e** Evolution of the magnetic moment with the atomic number of vacancies, showing that the magnetic moments extend from localized magnetic moment around vacancies (spin–orbit polarons, SOPs) to flat band negative magnetism from Co$_3$Sn kagome layer. The colored curve is a visual guide to the trend in the evolution of the moment with increasing atomic number $N$. The error bars in (**e**) are determined by linear fitting as shown in bottom panel in (**a**), (**b**) and (**d**). The red circling arrows in (**e**) indicate the circulating current in kagome lattice. STM parameters: (**a**): $V_s = -250$ mV, $I_t = 500$ pA; (**b**) $V_s = -50$ mV, $I_t = 500$ pA; (**d**) $V_s = 20$ mV, $I_t = 500$ pA; $V_{mod} = 0.2$ mV; (**c**) $V_s = -400$ mV, $I_t = 50$ pA.

and creation at the S surface of nonmagnetic Ni$_3$In$_2$S$_2$ (Supplementary Fig. 17). In addition, the vacancy repair at S surface, in another way, is the creation of S adatoms at exposed Co$_3$Sn surfaces inside the large-sized vacancy. Through manipulation technique, the artificial S adatom-based nanostructures with elaborate geometry inside large-sized S vacancy have been constructed (Supplementary Fig. 18).

The atomically precise manipulation of atomic vacancies opens a platform for artificial sulfur vacancy with custom-designed geometries and their coupling with physical parameters such as magnetic moment, orbital and charges in Co$_3$Sn$_2$S$_2$ and derivatives. In addition, the controlled integration of individual vacancies into extended, scalable atomic circuits with custom-designed shape and size, which may be promising for practical applications such as atomic memory[13] and quantum qubit[44]. Through precise engineering of vacancies, the artificial vacancy lattice can be achieved, which is essential for realizing designer quantum materials with tailored properties[14]. The couplings among the

vacancies in Co$_3$Sn$_2$S$_2$ could improve the understanding of polaron-like bound states in coupled quantum systems[42], to explore artificial coupled quantum systems with great control. The intriguing evolution of the magnetic moment with increasing vacancy size poses a challenge for theoretical frameworks supporting quantitative model calculations to understand many-body spin–orbit impurities[34,35] and negative flat band magnetism[43] in kagome magnet.

## Methods

### Single crystal growth of Co$_3$Sn$_2$S$_2$

The single crystals of Co$_3$Sn$_2$S$_2$ were grown by flux method with Sn/Pb mixed flux. The starting materials of Co (99.95% Alfa), Sn (99.999% Alfa), S (99.999% Alfa) and Pb (99.999% Alfa) arewere mixed in molar ratio of Co : S : Sn : Pb = 12 : 8 : 35 : 45. The mixture was placed in Al$_2$O$_3$ crucible sealed in a quartz tube. The quartz tube was slowly heated to 673 K over 6 h and kept over 6 h to avoid the heavy loss of sulfur. The

quartz tube was further heated to 1323 K over 6 h and kept for 6 h. Then the melt was cooled down slowly to 973 K over 70 h. At 973 K, the flux was removed by rapid decanting and subsequent spinning in a centrifuge. The hexagonal-plate single crystals with diameters of 2 ~ 5 mm are obtained. The composition and phase structure of the crystals were checked by energy-dispersive x-ray spectroscopy and x-ray diffraction, respectively.

### Scanning tunneling microscopy/spectroscopy
The samples used in the experiments were cleaved in situ at 6 K and immediately transferred to an STM head. Experiments were performed in an ultrahigh vacuum ($1 \times 10^{-10}$ mbar) ultra-low temperature STM system (40 mK) equipped with 9-2-2 T magnetic field. All the scanning parameter (setpoint voltage and current) of the STM topographic images are listed in the captions of the figures. Unless otherwise noted, the differential conductance (d$I$/d$V$) spectra were acquired by a standard lock-in amplifier at a modulation frequency of 973.1 Hz. Non-magnetic tungsten tip was fabricated via electrochemical etching and calibrated on a clean Au(111) surface prepared by repeated cycles of sputtering with argon ions and annealing at 500 °C.

### Vacancy manipulation
Through hundreds times attempts at surfaces of 3 $Co_3Sn_2S_2$ samples, the success rate of the S vacancy repair and creation in experiments is about 30% and 5%, respectively. The success rate depends on many conditions such as the pulse position, the pulse voltage and the sharpness of tip. The sharpness of tip is interpreted by the spatial resolution of STM topography. Normally, we get a higher success rate when putting on the sharper tip exactly on the center of vacancy and applying the same tip pulse voltage.

### Spin−polarized scanning tunneling microscopy/spectroscopy
Ferromagnetic Ni tip was applied in the spin−polarized STM measurement. The Ni tip was fabricated via electrochemical etching of Ni wire in a constant-current mode. To calibrate the spin-polarization of Ni tip, the as-prepared Ni tip has been applied to resolve magnetic-state dependent contrast of Co islands grown on a Cu(111) surface in spin−polarized STM experiments[45]. The two oppositely magnetized tips are achieved by applying a small magnetic field $B_z = 0.2$ T which is smaller than the coercivity of bulk sample (0.5 T, Supplementary Fig. 19) to solely flip the magnetization of tip but keep the magnetization of sample unchanged.

### Q-Plus nc-AFM measurements
Non-contact AFM measurements were performed on a combined nc-AFM/STM system (*Createc*) at 4.7 K with a base pressure lower than $2 \times 10^{-10}$ mbar. All measurements were performed using a commercial qPlus tuning fork sensor in the frequency modulation mode with a Pt/Ir tip at 4.5 K. The resonance frequency of the AFM tuning fork is 27.9 kHz, and the stiffness is approximately 1800 N/m.

### Model Hamiltonian
DFT calculations indicate the vacancy-vacancy forms vacancy states and contribute the peak DOS observed in experiment. Therefore, we simulated vacancies with a simple tight-binding model with a nearest neighbor hopping $t$. We constructed four vacancy patterns (linear, bi-linear, triangular, and hexagonal) with different number of S vacancies. In each vacancy configuration, the highest energy level is extracted as the vacancy state.

### First-principles calculations
The Nudged Elastic Band (NEB) calculations for the surfaces are simulated with a slab model of 4×4 in-plane supercell and four kagome layers along the out-of-plane direction. The vacuum level is about 14 Å. The NEB calculations were performed within Density Functional theory as implemented in VASP[46,47]. The generalized gradient approximation

parametrized by Perdew-Burke-Ernzerhof[48] is used to mimic the exchange-correlation interaction between electrons. A kinetic energy cutoff of 268 eV is used for the plane wave basis set. Single Gamma point is employed to sample the Brillouin zone. In the NEB structural relaxation, the force and total energy thresholds are about $10^{-4}$ eV and 0.01 eV/Å, respectively. The spring constant of 5 eV/Å$^2$ between neighboring images is used.

## Data availability
All data that support the findings of this study are present in the paper and the Supplementary Information. Further information can be acquired from the corresponding authors upon request.

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

## Acknowledgements

We thank Chendong Zhang and Claudia Felser for useful discussions. We thank Senhao Lv for assistance with the magnetization measurements. The work is supported by grants from the National Natural Science Foundation of China (61888102 (H.-J.G.) and 52022105(H.C.)), the National Key Research and Development Projects of China (2022YFA1204100 (H.Y. and H.C.), 2019YFA0308500 (H.-J.G. and L.H.), and 2018YFA0305800 (L.H.)), and the Chinese Academy of Sciences (YSBR-003 (H.C., L.H.)). Z. W. is supported by the US DOE, Basic Energy Sciences Grant No. DE-FG02-99ER45747and by Research Corporation for Science Advancement Cottrell SEED Award No. 27856. B.Y. was supported by financial support by the European Research Council (ERC Consolidator Grant "NonlinearTopo", No. 815869) and ISF -Singapore-Israel Research Grant (No. 3520/20).

## Author contributions

H.C. and H.-J.G. designed the experiments. H.C. and Y.-Q.X. performed STM experiments with guidance of H.-J.G. H.C., Y.-Q.X., B. H., Z.H. Y.Y. and X. H. analyzed experimental data, plotted figures. L.H., Q. Z. Y.L., and Y.X. performed AFM measurements. E.L., H.Y, X.Q., and H.L. prepared samples. H.Y. perform magnetization measurements. Z.W. proposed the model and H. T. and B. Y. carried out theoretical calculations. H.C. wrote the manuscript with the inputs from all authors. H.-J. G. supervised the project.

## Competing interests

The authors declare no competing interests.
