## [Peer Review File · Nature Communications]

Atomically precise engineering of spin–orbit polarons in a kagome magnetic Weyl semimetalREVIEWER COMMENTS

Reviewer #1 (Remarks to the Author):

Chen et al. report on scanning tunneling microscopy investigations of Sulfur vacancy sites on the surface of the magnetic Weyl semimetal $\text{Co}_3\text{Sn}_2\text{S}_2$. The authors present two main aspects. The first one is the controlled shaping of S-vacancy sites into surface structures of different size and shape. Second, those artificially produced defect sites are investigated for their defect states showing up in dI/dV spectroscopy with particular emphasis on the unusual magnetic behavior in an external magnetic field. The presented work is essentially a continuation of a previously published investigation of the same group (Ref. 35) with a new focus on the shape and size dependence of the defects discussed in Ref. 35 before. Both aspects of the presented investigation are interesting. However, I am missing a few details as well as deeper insights to be able to recommend the publication of the manuscript. Please consider my comments below for the preparation of a revised manuscript.

1) The authors nicely describe that they can modify the arrangement of S vacancy sites on the surface by bias pulses, i.e., they can deliberately close S-vacancy sites, which they explain by pulling up S atoms from sub-surface layers. As unlikely as this sounds, the presented calculated energy barriers appear reasonable for this process. I see, however, two issues here. Apparently, the vacancies create some localized states which is the main focus of the second part, but pulling up S atoms from inner layers does create vacancies inside the crystal and there is no word about that those vacancies do not (or do?) influence the vacancies on the top layer. It can be expected that there are correlations and thus I am not sure about the results presented later in the manuscript that are entirely discussed only on the basis of being a top surface defect.

To rectify this issue somewhat, I suggest to look into surface defects (without the so-called vacancy repair) that have the same shape as other defects created with the S-atom filling method and compare them with the artificial structures. This relies on the statistical appearance of certain shapes but looking at Figure S2 there is a high natural abundance of chain structures and even small triangles to look at. If one claims atomically engineering of defect structures, one should not forget that there is also a vacancy created inside the bulk.

This brings me to the second issue. The manuscript puts quite some emphasis on „atomically-precise construction of Sulfur quantum antidots“. It presents this as being an important issue to create „new artificial quantum materials“ with „exotic properties“. However, the creation, no matter how curious this is, is in my view very special to this material. This is by no means a general method to be used in any other bulk or large scale material of any other type. Hence, I doubt that this opens a „new playground“ as the generality of the method is not proven (or practically useful in the synthesis of such materials). Therefore, I would not present it as such a general method.

2) The study on the size and shape dependence of the magnetic state at the vacancy sites leaves me somewhat puzzled for several reasons. There is apparently some dependence, which is not too unexpected, but it leaves one without a greater picture of conclusions to be drawn. What has been

understood by this study, what new insights were gained besides the dependence of apparent magnetic moment on the size/shape? Can this be used to learn about the nature of the localized state? The tuneability of the magnetic state alone is no real step forward to me. It is also neither a demonstration of creating designer quantum states nor does it help in the understanding of the polaron-like state. If the authors can obtain some deeper insight from the data, it would make the manuscript a real advancement in the field.

I also cannot fully follow the origin of the magnetic state at the S vacancies. This was actually the topic of their previous article (Ref. 35), but the discussion reappears here. The material is a magnetic Weyl semimetal, thus I believe that the magnetic origin of the vacancy state is connected inherently to the magnetic state of the bulk. The authors present a magnetic signal for the vacancy state at zero field by using two oppositely magnetized tips of the STM. I assume that the surface state is considered to be magnetized equally for both tip magnetization directions. I further assume that to change the tip magnetization direction, the external field is ramped in opposite directions to create those tip states. The question is how the vacancy state behaves in these fields to switch the Ni tip. Does it stay the same or does it flip? For the magnetic field dependence measurement of the vacancy state the field is ramped in opposite directions and to explain the positive shift of the energy of the vacancy state its magnetic moment should flip with the external field easily as stated by the authors. Hence, I do not see how to get a magnetic signal for two opposite tip magnetizations if the vacancy state should have flipped as well. Therefore, I ask the authors to consider experimental conditions (i.e. B fields) at which the vacancy state flips (accordingly to the coercivity of the $\text{Co}_3\text{Sn}_2\text{S}_2$ - need to show also data for the bulk sample) and the coercivity of the Ni tip. This should allow to control the orientation of vacancy moment and tip magnetization independently. These B fields are relatively small but should have an impact on the dI/dV spectra.

Could the authors please clarify if the magnetic states of the vacancy are doublets or only single states locked to the bulk magnetic properties or due to the chiral nature of the surface state (I wonder then how the moment can be flipped). The evidence that these are not doublet states is due to the absence of a splitting of the states in high magnetic fields?

3) The presentation and discussion of the vacancy state is somewhat complex. The authors talk about anti-bonding state interaction, antidot quantum states etc. I think that the presentation can be simplified. It is clear that a vacancy creates a localized state. If two vacancies get close to each other, those states will interact like between two atoms. Quite some time ago, there were some STM studies on the evolution of emerging states when building up atomic structures atom by atom. Those similarities can be drawn here as well. Therefore, I suggest to present some sketch showing the creation of the potential in the valence band by the S vacancy and the emergence of localized states (from the spin-polarized surface states?). Bringing more vacancies together might be viewed as some sort of quantum corral structure if that picture is correct that leads to new bound states according to the new boundary conditions. If such a picture is valid, the authors could simplify their presentation since it is not clear what is meant by „anti-bonding interaction“. This is not a commonly used term. I also suggest to provide dI/dV maps of the larger shape to check the hypothesis on the evolution of quantum well states.

4) I also see some problems in the wording and descriptions used. Examples are „Kagome magnetic Weyl semimetal“ It suffices to say magnetic Weyl semimetal having a Kagome lattice but repeating the Kagome lattice throughout the manuscript does not add information. Further „defect-localized bound

states“, bound state fully suffices it is inherent that they are localized at the defect. „Quantum bound state“ - bound state. „S vacancy antidot“ - very unusual, just say S vacancy. Same for „S vacancy quantum antidot“. Also, what is „flat band negative magnetism“? A simpler language not emphasizing in every sentence „quantum“ and „antidot“ etc. would make the reading much easier. Is „spin-orbit polaron“ somewhere defined? Why is it polaronic in nature if the magnetic moment is due to a bound state? Does it polarize its surrounding and how is this proven?

5) I am missing almost all crucial experimental parameters in the manuscript. What are the conditions (temperature...) and setpoints used in the presented STM images and STS data acquisition.

Reviewer #2 (Remarks to the Author):

In this manuscript, the authors demonstrated experimentally the capability that S vacancies in a kagome magnetic Weyl semimetal of $\text{Co}_3\text{Sn}_2\text{S}_2$ can be controllably repaired using a properly pulsed STM tip. Enabled by this capability, the authors further studied the electronic and magnetic properties of vacancy antidots of varying sizes and shapes, and the main findings are interpreted within the picture of spin-orbital polarons. Beyond experimental observations, the authors also supported their main claims using modeling and simulations of different scales including first- principles calculations and tight-binding models.

Overall, the subject matter and system of study are of timely importance, and the vacancy repairing or manipulation capability the team demonstrated is impressive. The findings enabled by the capability are somewhat routine, lacking spectacular features, but as the authors conjectured in the concluding paragraph, more profound outcomes might indeed be expected based on the repairing mechanism demonstrated here. I therefore is in favor of paper's acceptance in Nature Communications. Nevertheless, the following questions and issues should be properly addressed first.

* Can the authors provide more convincing or direct evidence that the S_{up} needed to repair a single vacancy is from an S_{down} ? Can some distant S migrating on the top layer be absorbed into the vacancy at the pulse?

* Is the repairing process reversible, or irreversible? Will the polarity of the pulse have an effect on the repair or creation of an S vacancy?

* If the repairing process is NOT reversible, then the approach faces a severe limitation, in that one essentially has to work on the longest or largest vacancy antidots in order to study the size effects. Is this correct?

* A further related issue: can the pulse approach fuse two antidots into one, with each of the initial sizes larger than 1?

- * In Fig. 2c, is spin up or down absolutely defined?
- * Some more elaborations on the SOP would be desirable.
- * Some cross checking on the language would be necessary (e.g., In spired).

Reviewer #3 (Remarks to the Author):

Chen et. al. study the manipulation of sulfur vacancies on the surface of $\text{Co}_3\text{Sn}_2\text{S}_2$, a magnetic material with a Kagome magnetic lattice structure. The naturally occurring sulfur vacancies were repaired on the surface use voltage pulses from the tip of a scanning tunnelling microscope, which they propose occurs via a sulfur transition between a lower lying atomic layer and the surface. After demonstrating the successful manipulation of these single vacancies, the authors study vacancy pairs electronically and magnetically to understand the nature of the vacancy interactions. The authors then move on to construct larger vacancy structures, including single and double wide chains, triangles, and hexagons. They studied the electronic characteristics of each vacancy structure, revealing that the lowest energy bound state attributed to the vacancy is shifted closer to the Fermi energy with increasing vacancy island size. The authors finally study the magnetic response of the primary island bound state peak energy to estimate the local magnetic moment per vacancy as the island size changes.

While this is not the first demonstration of artificially constructed atomic ensembles on $\text{Co}_3\text{Sn}_2\text{S}_2$ (ref. 34 in the manuscript), this paper does shed new light on the magnetic characteristics of the localized vacancy states and how these states merge into the Co_3Sn bulk. While this finding is a significant development to understand the atomic origins of the negative flat band magnetism, at present, I think there are several significant issues with the manuscript that should be resolved.

1. It is unclear to me why these vacancies should be considered polarons. Local atomic relaxations of surface atoms around vacancies occur in a variety of materials, and are generally not considered polarons. That nomenclature is commonly reserved for systems where one internal degree of freedom (such as a charge) couples to a lattice distortion to form a many-body state. Furthermore, in those systems, the relaxation is not confined to the surface atomic layer – do the authors have evidence that the relaxations persist below the topmost atomic layer?
2. The picture of how the vacancies interact is completely unclear. Fig. 2a,b show that the electronic properties of the vacancies are essentially unchanged until the separation is $2a$ (this also rules out polaronic effects, as the lateral extension of the polarons should mediate interactions beyond nearest-neighbor distances), when a dramatic shift of the lowest energy peak occurs. Yet, even in Fig. S7 no clear indication of a bonding and anti-bonding peak is visible (none of the states have a nodal plane between

the vacancies), pointing to a more complex interaction than simple chemistry based on orbital overlap. For me, this nullifies the data from fig. 3j based on a simple tight-binding model. Also, it seems critical to address the later magnetic observations in fig. 4.

3. I find all data sets showing strain to be extremely unconvincing. The strain data shown in fig. 2 shows nearly all of the intensity in the vacancy region, where the strain would be ill-defined. Second, in fig. 2e and S11, the strain profile of the measured single vacancies varies widely and often appears to simply follow the background. If these arguments are going to be included in the paper, a detailed and thorough analysis of the single and double vacancies must be presented including the quantitative analysis of the strain characteristics for at least 10 members from each type (more would be better). In this way, it would be clear if the strain is statistically significant from the background. This analysis would also have to confirm that the measurements are not influenced by the magnitude of the chosen applied bias.

4. The authors only mention the possibility of repairing existing vacancies, not the ability to define new ones. Does this mean the scale of the structures is limited to the size of the vacancy islands which are naturally occurring? I think this would be a severe limitation for potentially scaling this system beyond the work shown here.

5. Spin-sensitive detection of electronic states is difficult because of its convolution with the electronic states – this is particularly true in determining the stabilisation conditions when measuring magnetic systems. In principle, to compare two magnetic measurements, they measurements have to be taken using a stabilisation condition where it is known that no magnetic contrast is present. As the authors don't discuss the stabilisation strategy at all, the utility of the measurements in fig. 2c is called into question. Also, the yellow curves in the bottom panel are completely indistinguishable.

6. How are the peak positions for the bound states determined (e.g. fig. 2d, but also for fig. 4)? It seems quite a challenge to me because the peaks are clearly not simple Gaussians or Lorentzians, in fact some look as though they are more than one peak, and it actually seems that most of the curves in fig. 2d actually have a similar onset around -225 meV. It also looks as though the peak width (or perhaps one of the internal peaks) is changing in the magnetic field. Have the authors tried fitting these peaks to determine (1) if they can be fit with single Gaussian or Lorentzian peaks, (2) if the peak widths are changing, and (3) if there are multiple peaks in these ground states. This level of detail would certainly aid in understanding the "shifts" seen with magnetic field and should be examined in more detail.

7. The negative magnetic moments in this material are ascribed to the orbital magnetization – can the authors explain why this orbital moment would decrease for a single vacancy? Additionally, why should this moment be so different from substitutional In impurities, which show an effective moment of $-5\mu_B$.

Additional technical questions:

What is the success rate of vacancy repair? Does it depend on tip state? What is the tip conditioning prior to manipulation?

The proposed mechanism of sulfur migration would leave a sulfur vacancy in the subsurface layer, it seems surprising that this would not have an electronic effect on the surface properties, have the authors calculated the LDOS for one such sulfur subsurface vacancy?

Also, is it possible that sulfur is transferred from the tip to the surface? Can the authors rule this out?

Where are the spectra in fig. 2b taken?

Why is the peak spacing strongly modified in the case of the dimer?

What direction is up in fig. 2c?

Why do the peaks above (although they are lower in absolute energy) the lowest energy bound state change so strongly with varying island/structure size?

What is the size of the error bars in fig. 3i? How many measurements are taken for each data point?

How are the error bars in fig. 4e determined? How many measurements are taken per point here?

Point-by-point response to the comments from the reviewers

Response to Reviewer 1[#]

Overall Critical Comment:

“Chen et al. report on scanning tunneling microscopy investigations of Sulfur vacancy sites on the surface of the magnetic Weyl semimetal $\text{Co}_3\text{Sn}_2\text{S}_2$. The authors present two main aspects. The first one is the controlled shaping of S-vacancy sites into surface structures of different size and shape. Second, those artificially produced defect sites are investigated for their defect states showing up in dI/dV spectroscopy with particular emphasis on the unusual magnetic behavior in an external magnetic field. The presented work is essentially a continuation of a previously published investigation of the same group (Ref. 35) with a new focus on the shape and size dependence of the defects discussed in Ref. 35 before. Both aspects of the presented investigation are interesting. However, I am missing a few details as well as deeper insights to be able to recommend the publication of the manuscript. Please consider my comments below for the preparation of a revised manuscript.”

Response: We thank the reviewer for the comments and suggestions. In the following, we have taken reviewer’s suggestions, added the details and made corresponding revisions in the revised manuscript to address all of the concerns from the reviewer.

Comment 1. *The authors nicely describe that they can modify the arrangement of S vacancy sites on the surface by bias pulses, i.e., they can deliberately close S-vacancy sites, which they explain by pulling up S atoms from sub-surface layers. As unlikely as this sounds, the presented calculated energy barriers appear reasonable for this process. I see, however, two issues here.*

Issue 1. *Apparently, the vacancies create some localized states which is the main focus of the second part, but pulling up S atoms from inner layers does create vacancies inside the crystal and there is no word about that those vacancies do not (or do?) influence the vacancies on the top layer. It can be expected that there are correlations and thus I am not sure about the results presented later in the manuscript that are entirely discussed only on the basis of being a top surface defect.*

To rectify this issue somewhat, I suggest to look into surface defects (without the so-called vacancy repair) that have the same shape as other defects created with the S-atom filling method and compare them with the artificial structures. This relies on the statistical appearance of certain shapes but looking at Figure S2 there is a high natural

abundance of chain structures and even small triangles to look at. If one claims atomically engineering of defect structures, one should not forget that there is also a vacancy created inside the bulk.

Response 1-1. We agree with the Reviewer that the discussion of creating vacancies inside the crystal in the vacancy repair process is necessary. We took the Reviewer’s suggestions and studied the naturally-formed surface defects with same geometry with as-created vacancies (new Fig. S6). We found that the localized bound states around these naturally-formed defects are similar with the artificially defects, which indicates that the vacancy created inside the crystal do not make considerable contribution to the density of states of the top surface vacancies near Fermi level.

Fig. S6. Comparison of bound states between various artificially-created vacancy and naturally-formed vacancies. **a,b** STM images showing two cases of repairing a dimer vacancy into a single vacancy. ($V_s = -400$ mV, $I_t = 500$ pA). **c**, The dI/dV spectra obtained at artificial single vacancies in **(a)** and **(b)**, showing similar features with the naturally-formed single vacancies near Fermi level. ($V_s = -400$ mV, $I_t = 500$ pA, $V_{\text{mod}} = 0.5$ mV).

Accordingly, we have added these new data in Fig. S6 and added discussions about the vacancy created inside the bulk in Page 7 in the revised manuscript.

“Pulling up S atoms from inner layer will create vacancies inside the crystal. These vacancies do not make observable contribution to the density of states at top surface states near Fermi level. The dI/dV spectra between artificially-created and naturally-formed vacancies with the same shape on the top S layer show similar lineshapes near Fermi level (Fig. S6).”

Issue 2. This brings me to the second issue. The manuscript puts quite some emphasis on “atomically-precise construction of Sulfur quantum antidots”. It presents this as

being an important issue to create “new artificial quantum materials” with “exotic properties”. However, the creation, no matter how curious this is, is in my view very special to this material. This is by no means a general method to be used in any other bulk or large scale material of any other type. Hence, I doubt that this opens a “new playground” as the generality of the method is not proven (or practically useful in the synthesis of such materials). Therefore, I would not present it as such a general method.

Response 1-2. We agree that the method is not general to all materials but limited in the specific material system because it depends on many conditions such as crystalline structures and the energy barrier for the interlayer migration of surface atoms.

To further demonstrate that the engineering of vacancies provide a new platform for the formation of artificial sulfur vacancy/adatom based nanostructures, we would like show the new experimental results as follows:

(1) The vacancy manipulation method is not limited to ferromagnetic $\text{Co}_3\text{Sn}_2\text{S}_2$ but can extend to other $\text{Co}_3\text{Sn}_2\text{S}_2$ -derived kagome metals. We have shown one exemplary material $\text{Ni}_3\text{In}_2\text{S}_2$ which is nonmagnetic (see *npj Comput. Mater.* **8**, 155 (2022)), as shown in the following Figure S17.

Figure S17. Vacancy manipulation in a $\text{Co}_3\text{Sn}_2\text{S}_2$ -derived kagome metal, $\text{Ni}_3\text{In}_2\text{S}_2$. **a**, Atomic structure of $\text{Ni}_3\text{In}_2\text{S}_2$. **b**, STM images showing the topography before (left) and after (right) vacancy creation, demonstrating the remove of S atom at surface. **c**, STM images showing the topography before (left) and after (right) vacancy repair, demonstrating the filling of S atom. STM scanning parameters for (b-c): $V_s = -300$ mV; $I_t = 500$ pA.

(2) The vacancy repair at S surface, in another way, is the creation of S adatoms at exposed Co_3Sn surfaces inside the large-sized vacancy (Fig. S18 (a)). Through

similar manipulation technique, the artificial S adatom-based nanostructures with elaborate geometry inside large-sized S vacancy have been formed (Fig. S18 (b-d)).

Fig. S18. Engineering of the S atoms inside large-size vacancy. **a**, Schematics of tip-assisted atomically-precise engineering of S atoms inside large-size S vacancy. **b**, STM image showing that exposure of Co₃Sn layer when the S vacancies at S-terminated surface become relatively large. Some single S adatoms highlighted by the white arrows are observed. **c**, STM images showing the construction of S atomic chains with gradually increased length. **d**, Series of STM images showing the process to write a “S” letter by engineering the atomic S adatoms. The scale bar in all figures are 1 nm.

Accordingly, we have revised the manuscript to replace the expressions on general method “new playground” “atomically-precise construction of Sulfur quantum antidots” “new artificial quantum materials with exotic properties” with detailed words “new platform” “formation”, “artificial sulfur vacancy” in the abstract and main text of the revised manuscript.

Abstract: “We achieve the step-by-step repairing of the selected vacancies, which results in the formation of artificial sulfur vacancy with elaborate geometry.”

Discussion part of main text in page 13: “The atomically-precise engineering of sulfur vacancies is not limited to ferromagnetic Co₃Sn₂S₂ but can extend to other Co₃Sn₂S₂-derived kagome metals. We have also achieved the controlled vacancy repair and creation at the S surface of nonmagnetic Ni₃In₂S₂ (Fig. S17). In addition, the vacancy repair at S surface, in another way, is the creation of S adatoms at exposed Co₃Sn surfaces inside the large-sized vacancy. Through similar manipulation technique, the

artificial S adatom-based nanostructures with elaborate geometry inside large-sized S vacancy have been formed (Fig. S18).”

“The atomically-precise manipulation of atomic vacancies opens a new platform for artificial sulfur vacancy with custom-designed geometries and their coupling with physical parameters such as magnetic moment, orbital and charges in $\text{Co}_3\text{Sn}_2\text{S}_2$ and derivatives.”

Comment 2-1. The study on the size and shape dependence of the magnetic state at the vacancy sites leaves me somewhat puzzled for several reasons. There is apparently some dependence, which is not too unexpected, but it leaves one without a greater picture of conclusions to be drawn. (1) What has been understood by this study, (2) what new insights were gained besides the dependence of apparent magnetic moment on the size/shape? (3) Can this be used to learn about the nature of the localized state? The tuneability of the magnetic state alone is no real step forward to me. It is also neither a demonstration of creating designer quantum states nor does it help in the understanding of the polaron-like state. If the authors can obtain some deeper insight from the data, it would make the manuscript a real advancement in the field.

Response 2-1: In our previous study of single vacancy (Ref. 35), the bound states show anomalous Zeeman effect with a local magnetic moment of about $1.35\mu_B$. Meantime, the negative flat band magnetism of the bulk states in $\text{Co}_3\text{Sn}_2\text{S}_2$ which is due to the orbital contribution in the flat band portion of the itinerant electrons. The magnetic moment of itinerant electrons is about $3\mu_B$ (Ref. 43, *Nat. Phys.* 15, 443 (2019)). Therefore, the interesting question is what the evolution of magnetic moment from local electrons to itinerant electrons in this flat band magnetism system. The natural idea for addressing such question is extending the single vacancy to large-sized exposed kagome surface. The expanding of vacancy size dependent on the vacancy shape by considering the lattice symmetry and inter-vacancy hybridization. Thus, the motivation for the size and shape dependence of magnetic states at the vacancy site is to shed new light on the magnetic characteristics of the localized vacancy states and how these states merge into the Co_3Sn bulk, as correctly pointed by the Reviewer 3[#] in his/her report. It is very important for understanding the interaction of vacancy bound states and connection to the negative magnetism from kagome flat band. Furthermore, for practical application, shape and size effects have to be taken into account in designing two-dimensional arrays of magnetic nanostructures for potential application based on spin-orbit polarons.

Accordingly, we have revised the sentences to highlight the motivations.

“To gain deep understanding of nature of localized states at vacancies, we further study the shape-dependent magnetic moment of the bound states. We focus on ...”

“In addition, the controlled integration of individual vacancies into extended, scalable atomic circuits with custom-designed shape and size, which is promising for practical applications such as atomic memory¹³ and quantum qubit”

Comment 2-2. I also cannot fully follow the origin of the magnetic state at the S vacancies. This was actually the topic of their previous article (Ref. 35), but the discussion reappears here. The material is a magnetic Weyl semimetal, thus I believe that the magnetic origin of the vacancy state is connected inherently to the magnetic state of the bulk. The authors present a magnetic signal for the vacancy state at zero field by using two oppositely magnetized tips of the STM. I assume that the surface state is considered be magnetized equally for both tip magnetization directions. I further assume that to change the tip magnetization direction, the external field is ramped in opposite directions to create those tip states. The question is how the vacancy state behaves in these fields to switch the Ni tip. Does it stay the same or does it flip? For the magnetic field dependence measurement of the vacancy state the field is ramped in opposite directions and to explain the positive shift of in energy of the vacancy state its magnetic moment should flip with the external field easily as stated by the authors. Hence, I do not see how to get a magnetic signal for two opposite tip magnetizations if the vacancy state should have flipped as well. Therefore, I ask the authors to consider experimental conditions (i.e. B fields) at which the vacancy state flips (accordingly to the coercivity of the Co₃Sn₂S₂ - need to show also data for the bulk sample) and the coercivity of the Ni tip. This should allow to control the orientation of vacancy moment and tip magnetization independently.

Response 2-2: We thank the Reviewer for the comments and suggestions on the effect of magnetic field on the vacancy states when measuring the spin-polarized STM. Before the spin-polarized STM measurements, we have measured the bulk magnetization curve of Co₃Sn₂S₂ with magnetic field applied along *c* axis and obtained the coercivity of the Co₃Sn₂S₂ (about 0.5 T, Fig. R1). In the spin-polarized STM measurements of this work, external magnetic field with small magnitude (0.2 T, small than the coercivity of the Co₃Sn₂S₂, ~0.5 T) is applied to switch the magnetization of Ni tip. At such small magnetic field, the tip magnetization is switched independent of the orientation of vacancy moment.

Figure R1. Bulk magnetization curve at 2 K with out-of-plane magnetic field

Accordingly, we added more details in the Method part at Page 18 of revised manuscript. In addition, we have added a simple schematic inside Fig. 2c to show the magnetization of the tip and sample during the spin-polarized STM measurements.

Revised Fig. 2c. Inset: schematics showing the magnetization of tip and the vacancy bound state.

“Thus, we further study the magnetic properties of the bound states around dimer vacancy through spin-polarized STM (details see Method).”

“The two oppositely magnetized tips are achieved by applying a small magnetic field $B_z=0.2$ T which is smaller than the coercivity of bulk sample (0.5 T) to solely flip the magnetization of tip but keep the magnetization of sample unchanged.”

Comment 2-3. These B fields are relatively small but should have an impact on the dI/dV spectra. Could the authors please clarify if the magnetic states of the vacancy are doublets or only single states locked to the bulk magnetic properties or due to the chiral nature of the surface state (I wonder then how the moment can be flipped). The evidence

that these are not doublet states is due to the absence of a splitting of the states in high magnetic fields?

Response 2-3: Yes, due to the absence of a splitting in high magnetic field, the magnetic states of the vacancy are single states. Such single state is mainly originated from the diamagnetic orbital current contribution and pseudospin (electron spin + atomic orbital) in the presence of spin-orbit coupling. In that case, the single states exhibits a diamagnetic behavior where the net magnetization is always oriented antiparallel to the direction of the magnetic field (details see our previous work Ref. [35], Nature Communications **11**, 4415 (2020)).

***Comment 3.** The presentation and discussion of the vacancy state is somewhat complex. The authors talk about anti-bonding state interaction, antidot quantum states etc. I think that the presentation can be simplified. It is clear that a vacancy creates localized state. If two vacancies get close to each other, those states will interact like between two atoms. Quite some time ago, there were some STM studies on the evolution of emerging states when building up atomic structures atom by atom. Those similarities can be drawn here as well. Therefore, I suggest to present some sketch showing the creation of the potential in the valence band by the S vacancy and the emergence of localized states (from the spin-polarized surface states?). Bringing more vacancies together might be viewed as some sort of quantum corral structure if that picture is correct that leads to new bound state according to the new boundary conditions. If such a picture is valid, the authors could simplify their presentation since it is not clear what is meant by “anti-bonding interaction”. This is not a commonly used term. I also suggest to provide dI/dV maps of the larger shape to check the hypothesis on the evolution of quantum well states.*

Response 3: We thank the reviewer for the comments. We have taken the reviewer’s suggestions and simplified the presentation by presenting a sketch (new Fig. 3j). The vacancy bound states are not originated from the spin-polarized surface states (the surface states at S surface is not spin-polarized as demonstrated in our previous work in Ref. [35], Nature Communications **11**, 4415 (2020)). These bound states are originated from the localization of the Co *d* electrons by the vacancy potential of the missing S²⁻ ion and diamagnetic orbital current are trapped by the lattice distortion around the vacancies. Although the origin of bound states is not as simple as the normal vacancy states simply induced by the vacancy potential, the tight binding model in Figure 3k is indeed based on the simple picture pointed by the reviewer, and indicates that quantum confinement effect of bound states.

Accordingly, we added the sketch as a new Fig. 3j and corresponding discussions as follow.

“We simulate vacancies with a simple tight-binding model with a nearest neighbor hopping t by considering the hybridization between vacancies (Fig. 3j).”

New Fig. 3j. Schematic illustration of hybridization between vacancy bound states. Single vacancy creates bound states ($|\Psi|^2$) inside the gap-like density of states at S surface. The hybridization of the additional vacancy induces a new bound state at higher energy.

According to the reviewer’s suggestion, we have collected dI/dV map of $N=28$ triangular vacancy. The primary peak with lowest energy (Fig. S13 (c)) is localized at the center of triangle while other lower energy peak gradually expand from triangular center to the edges (Fig. S13 (d,e)), indicating the quantum confinement effect.

Accordingly, we have deleted the “anti-bonding interaction” and added the discussions of quantum confinement as follows.

“Furthermore, the spatial distributions of the bound states across the single chain vacancies (Fig. S9) exhibit quasi one-dimensional band behaviors⁴⁰⁻⁴², indicating the existences of vacancy-vacancy interactions. The dI/dV maps at large-size triangular vacancy show quantum confinement effect, featuring a quantum antidot (Fig. S13)”

Fig. S13. dI/dV maps of large-size triangular S vacancies. **a**, STM topography of large-size triangular vacancy ($N=28$). **b**, dI/dV spectra obtained at the vacancy center (red triangle in **a**) and the S surface, respectively, showing the sharp conductance peak at -3.7 mV. **c-e**, the dI/dV maps at -3.7 mV (**c**), -50 mV (**d**) and -100 mV (**e**), respectively, showing the quantum confinement. STM parameter: $V_s = -400$ mV, $I_t = 500$ pA, $V_{\text{mod}} = 0.5$ mV.

Comment 4. I also see some problems in the wording and descriptions used. Examples are “Kagome magnetic Weyl semimetal”. It suffices to say magnetic Weyl semimetal having a Kagome lattice but repeating the Kagome lattice throughout the manuscript does not add information. Further “defect-localized bound states”, bound state fully suffices it is inherent that they are localized at the defect. “Quantum bound state” - bound state. “S vacancy antidote” - very unusual, just say S vacancy. Same for “S vacancy quantum antidote”. Also, what is “flat band negative magnetism”? A simpler language not emphasizing in every sentence “quantum” and “antidote” etc. would make the reading much easier. Is “spin-orbit polaron” somewhere defined? Why is it polaronic in nature if the magnetic moment is due to a bound state? Does it polarize its surrounding and how is this proven?

Response 4: We thank the reviewer for the comments. We have made revisions according to these comments as follows:

Wording and descriptions.

Except for the one or two sentences, we have taken the reviewer’s suggestions and revised the following wording in most sentences of revised manuscript to make the language simpler for read:

- (1) “kagome magnetic Weyl semimetal” into “magnetic Weyl semimetal”
- (2) “defect-localized bound states”- “bound states”
- (3) “quantum bound state”- “bound states”
- (4) “S vacancy antidot”-“S vacancy”
- (5) “flat band negative magnetism” means negative magnetic moments resulting from spin-orbit coupling in the kagome flat band as reported in Ref. 43 (Nature Physics **15**, 433 (2019)). To avoid confusion, we have reworded it as “negative magnetic moments resulting from spin-orbit coupling in the kagome flat band”.

Definition of “spin-orbit polarons”.

As the “spin-orbit polarons” have been defined and demonstrated in our previous work (Ref. 35, Nature Communications **11**, 4415 (2020)), we did not define it in this work but just cite the paper. As the reviewer concerns on the definition and the polaron nature, we would like to explain the concept as follows. Firstly, the spin-polarized STM measurements have demonstrated that, the S-terminated surface is not spin-polarized while **the vacancy bound state is spin-polarized**. In addition, the spatial distribution of bound states is directly correlated to the underlying kagome lattice. These results demonstrate that the magnetic moment of the bound state is not result from surface state projected at the S surface but from the localization of the Co *d* electrons by the vacancy potential of the missing S^{2-} ion, which hybridize with the S *p*-electrons. Secondly, the observed anomalous Zeeman response, the energy of bound state linearly shift to higher energy side independent of the direction of magnetic field, implicates that a **dominant diamagnetic orbital current contribution**. Based on above two conclusions, the polaron nature featured by the local spin and orbital current trapped by the lattice distortion around the vacancies due to the spin-orbit coupling. Normally, the spin-trapped polaron is named by magnetic polaron. To emphasize its orbital magnetic moment, we dub the magnetic bound state as a localized spin-orbit polaron (SOP).

We understand that the concept of “spin-orbit polaron” should be defined to avoid confusion.

According to the Reviewer’s comments, we have added the definition of SOP:

“The bound states around the single vacancy have been demonstrated to emergent from the SOP³⁵, where vacancy-potential localized spin from the Co *d* electrons and diamagnetic orbital current are trapped by the lattice distortion around the vacancies”.

Comment 5. I am missing almost all crucial experimental parameters in the manuscript. What are the conditions (temperature ...) and setpoints used in the presented STM images and STS data acquisition.

Response 5: We thank the Reviewer's comment and have added all the crucial experimental parameters in each Figure captions in the revised manuscript and supplementary information.

Figure 1: STM scanning parameters for **(b-c)**, **(e-f)** and **(h-i)**: Sample bias $V_s = -400$ mV; Current setpoint $I_t = 500$ pA.

Figure 2: (a) ($V_s = -400$ mV, $I_t = 500$ pA, $V_{\text{mod}} = 0.2$ mV).

(b) ($V_s = -360$ mV, $I_t = 500$ pA).

(c) ($V_s = -350$ mV, $I_t = 500$ pA)

(d) ($V_s = -300$ mV, $I_t = 2$ nA, $V_{\text{mod}} = 0.1$ mV)

Figure 3: STM parameters for **(a-h)**: $V_s = -400$ mV, $I_t = 500$ pA, $V_{\text{mod}} = 0.2$ mV.

Figure 4: STM parameter: (a): $V_s = -250$ mV, $I_t = 500$ pA; (b) $V_s = -50$ mV, $I_t = 500$ pA; (d) $V_s = 20$ mV, $I_t = 500$ pA; $V_{\text{mod}} = 0.2$ mV; (c) $V_s = -400$ mV, $I_t = 50$ pA.

Response to Reviewer 2[#]

In this manuscript, the authors demonstrated experimentally the capability that S vacancies in a kagome magnetic Weyl semimetal of $\text{Co}_3\text{Sn}_2\text{S}_2$ can be controllably repaired using a properly pulsed STM tip. Enabled by this capability, the authors further studied the electronic and magnetic properties of vacancy antidots of varying sizes and shapes, and the main findings are interpreted within the picture of spin-orbital polarons. Beyond experimental observations, the authors also supported their main claims using modeling and simulations of different scales including first-principles calculations and tight-binding models. Overall, the subject matter and system of study are of timely importance, and the vacancy repairing or manipulation capability the team demonstrated is impressive. The findings enabled by the capability are somewhat routine, lacking spectacular features, but as the authors conjectured in the concluding paragraph, more profound outcomes might indeed be expected based on the repairing mechanism demonstrated here. I therefore is in favor of paper's acceptance in Nature Communications. Nevertheless, the following questions and issues should be properly addressed first.

Response: We thank the reviewer for the positive remarks on our work. In the following, according to reviewer's comments and suggestions, we have done further revisions to address all of the concerns from the reviewer, and made the revisions correspondingly.

Comment 1. *Can the authors provide more convincing or direct evidence that the S_{up} needed to repair a single vacancy is from an S_{down} ? Can some distant S migrating on the top layer be absorbed into the vacancy at the pulse?*

Response 1: We thank the reviewer for the comments. We have provided more evidences for the vacancy repair mechanism.

(1) In the response to the next comments (Response 2), we have demonstrated that the manipulation of S vacancy is reversible and found that the success rate of vacancy creation is much lower than that of vacancy repair, which further supports the proposed mechanism. On one hand, if the S atoms are originated from the underlying S layer, the vacancy creation at top surface relies heavily on the identification of vacancies at underlying S layer which is difficult for surface-sensitive STM technique as the signals are mostly contributed from top S layer. As a result, the success rate should be much lower than the vacancy repair due to the easy identification of the top layer vacancies, which is consistent with the experiments. On the other hand, if the S atoms are originated from the adatoms at top surface or STM tip, the success rate for the vacancy creation and repair should be similar due to the close value of the energy barrier of S migration.

- (2) In the present work, we did not observe the migration of S adatoms at the largest field of view in STM ($800 \text{ nm} \times 800 \text{ nm}$) when applying a tip pulse at a low temperature of $<5 \text{ K}$. In addition, the barrier for the pushing or dragging the S adatoms by STM tip is so high that the manipulation of S atoms at top S surfaces have not been achieved in both this work and other STM works in the literature, which excludes the distant S migrating on the top layer be absorbed into the vacancy at the pulse.
- (3) We exclude that the additional S are originated from the STM tip because the vacancy repair is achieved by using a clean tip which is immediately transferred into the STM chamber once calibrated at Au surfaces.

Accordingly, we have added these discussions in the revised manuscript as follows.

“The origin of additional S atom to fill the single vacancy is illustrated in Figure 1d. There are no topographic changes between the relatively large scale STM images before and after the tip pulse except for the vacancy repair (Fig. S4). In addition, the diffusion energy for the migrating S atoms onto the S surface or STM tip is large due to the strong bonding between the surface S atoms and the underlying Co_3Sn layer. The manipulation is achieved by using a clean tip which is immediately transferred into the STM chamber once calibrated at Au surfaces. Therefore, it suggests that the filling S atom originates from the underlying layers rather than the surface or STM tip.”

“The lower success rate of vacancy creation further supports the proposed manipulation mechanism because the vacancy creation at top surface relies heavily on the existence of S vacancy around the creation position at the lower layer which is difficult to be identified by STM.”

Comment 2. *Is the repairing process reversible, or irreversible? Will the polarity of the pulse have an effect on the repair or creation of an S vacancy?*

Response 2: The repairing process is reversible, as shown in the following Figure. We did not find the effect of the pulse polarity. The success rate for the vacancy creation (5%) is much lower than the vacancy repair (30%) in experiments.

The repair or creation are independent of the polarity of the pulse. We have achieved the vacancy repair/creation at top S surface by applying both positive and negative pulse. Accordingly, we have added the creation of S vacancy in the revised Figure 1 and the corresponding discussions in the main text as follows.

“The vacancy repair at S surface is reversible (Fig. 1e). By putting the tip upon one of the upper S atoms (marked by the black cross in Fig. 1e) and applying a voltage pulse,

we have achieved the vacancy creation at S surface (Fig. 1f). The success rate for the vacancy creation (5%) is much lower than repair (30%) in experiments (details see Method). The lower success rate of vacancy creation further supports the proposed manipulation mechanism because the vacancy creation at top surface relies heavily on the existence of S vacancy around the creation position at the lower layer which is difficult to be identified by STM.”

“Briefly, to repair a single S vacancy, we position the STM tip close to the vacancy center (red arrow in Fig. 1b), followed by applying a tip pulse with a small voltage (approximately ranges from 0.5 V to 1.0 V, independent of the pulse polarity).”

Revised Fig. 1. e,f STM images showing the topography before (a) and after (b) vacancy creation, demonstrating the remove of S atom at surface. g, Schematic showing the remove of S atom from top S layer to fill the vacancy at the bottom S layer.

Comment 3. If the repairing process is NOT reversible, then the approach faces a severe limitation, in that one essentially has to work on the longest or largest vacancy antidots in order to study the size effects. Is this correct?

Response 3: No, the repairing process is reversible as shown in the Response 2. Thus, we did not face a severe limitation.

Comment 4. A further related issue: can the pulse approach fuse two antidots into one, with each of the initial sizes larger than 1?

Response 4: We thank the reviewer for the further related issue. The fusion of two vacancy into larger one is feasible but challenging in this work as the success rate of vacancy creation is relatively low. After continues attempts, we have achieved a simple fusion case where two dimer vacancies fuse into a V-shaped vacancy though step by

step vacancy repair (red arrow) and creation (black cross) as shown in the following Figure (Fig. R2).

Fig. R2. STM images before (a) and after (b) manipulation, respectively, showing the fusion of two dimer vacancies into a V-shaped vacancy. The red arrow indicates the position of tip pulse for the vacancy repair. The black cross indicates the position of tip pulse for the vacancy creation. The Scanning parameters: $V_s = -400$ mV, $I_t = 500$ pA.

Comment 5. In Fig. 2c, is spin up or down absolutely defined?

Response 5: We define the spin direction as “up” and “down” with respect to cleaved S surfaces for STM measurements.

To make the definition of spin clear, we have added the definition in the text and the schematics in Fig. 2c.

“We define the spin direction as “up” and “down” with respect to cleaved S surfaces.”

Revised Fig. 2c. Inset: schematics showing the magnetization of tip and the vacancy bound state.

Comment 6. Some more elaborations on the SOP would be desirable.

Response 6: We have adopted the reviewer’s suggestion and have added more elaborations on the SOP as follows.

- (1) In addition to the shape and size, the orientation of large vacancies with regular shape may also affect the electronic properties. For example, there are two configurations for the $N=3$ triangular vacancies. One is upward triangle which

means that the Co_3Sn kagome unit is centered in the triangular vacancy (In Fig. 3e and Fig. 4, the vacancies are all upward triangles). Another case is downward triangles where the hexagon of Co_3Sn kagome unit is centered in the triangular vacancy. The hybridization of the S p -electrons between $3d$ orbital at the underlying Co_3Sn layer is different for two cases, which results in the bound states at distinct energy.

Fig. S11. Comparison between upward and downward triangular vacancies. **a**, Schematic showing the upward triangular vacancy where a Co-based triangle of the Co_3Sn kagome unit is centered in vacancy. **b-d**, STM image, dI/dV map at -207 mV and dI/dV curves obtained at the position marked by white circle in **(b)**, respectively, showing the primary bound state is strongest at the vacancy center. **e**, Schematic showing the downward triangular vacancy where a hexagon of Co_3Sn kagome unit is centered in vacancy. **f-h**, STM image, dI/dV map at -231 mV and dI/dV curves obtained at the position marked by white circle in **(f)**, respectively, showing the distribution of primary bound state. The scale bars in all figures correspond to 1 nm. STM parameters: $V_s = -400$ mV, $I_t = 500$ pA, $V_{\text{mod}} = 0.2$ mV.

(3) The vacancy repair at S surface, in another way, is the creation of S adatoms at exposed Co_3Sn surfaces inside the large-sized vacancy (Fig. S17 (a)). Through similar manipulation technique, the artificial S adatom-based nanostructures with elaborate geometry inside large-sized S vacancy have been formed (Fig. S17 (b-d)).

Fig. S18. Engineering of the S atoms inside large-size vacancy. **a**, Schematics of tip-assisted atomically-precise engineering of S atoms inside large-size S vacancy. **b**, STM image showing that exposure of Co_3Sn layer when the S vacancies at S-terminated surface become relatively large. Some single S adatoms highlighted by the white arrows are observed. **c**, STM images showing the construction of S atomic chains with gradually increased length. **d**, Series of STM images showing the process to write a “S” letter by engineering the atomic S adatoms. The scale bar in all figures are 1 nm.

Accordingly, we have added these new results as additional supplementary Figures as new Fig. S11 and Fig. S18.

Comment 7. Some cross checking on the language would be necessary (e.g., In spired).

Response 7: Thanks for the suggestions. We have revised all the typos throughout the manuscripts.

“In spired by” have been revised into “Inspired by...”

“step-to-step” has been revised into “step-by-step”

“spacings” has been revised into “spacing”

“singlse” has been revised into “single”

Response to Reviewer 3[#]

Overall Critical Comment:

Chen et. al. study the manipulation of sulfur vacancies on the surface of $\text{Co}_3\text{Sn}_2\text{S}_2$, a magnetic material with a Kagome magnetic lattice structure. The naturally occurring sulfur vacancies were repaired on the surface use voltage pulses from the tip of a scanning tunnelling microscope, which they propose occurs via a sulfur transition between a lower lying atomic layer and the surface. After demonstrating the successful manipulation of these single vacancies, the authors study vacancy pairs electronically and magnetically to understand the nature of the vacancy interactions. The authors then move on to construct larger vacancy structures, including single and double wide chains, triangles, and hexagons. They studied the electronic characteristics of each vacancy structure, revealing that the lowest energy bound state attributed to the vacancy is shifted closer to the Fermi energy with increasing vacancy island size. The authors finally study the magnetic response of the primary island bound state peak energy to estimate the local magnetic moment per vacancy as the island size changes.

*While this is not the first demonstration of artificially constructed atomic ensembles on $\text{Co}_3\text{Sn}_2\text{S}_2$ (ref. 34 in the manuscript), *this paper does shed new light on the magnetic characteristics of the localized vacancy states and how these states merge into the Co_3Sn bulk. While this finding is a significant development to understand the atomic origins of the negative flat band magnetism, at present, I think there are several significant issues with the manuscript that should be resolved.**

Response: We thank the reviewer for the constructive comments and suggestions of our work. In the following, according to reviewer's comments and suggestions, we have done further revisions to address all of the concerns from the reviewer, and made the revisions correspondingly.

Comment 1. *It is unclear to me why these vacancies should be considered polarons. Local atomic relaxations of surface atoms around vacancies occur in a variety of materials, and are generally not considered polarons. That nomenclature is commonly reserved for systems where one internal degree of freedom (such as a charge) couples to a lattice distortion to form a many-body state. Furthermore, in those systems, the relaxation is not confined to the surface atomic layer – do the authors have evidence that the relaxations persist below the topmost atomic layer?*

Response 1. We thank the reviewer for the comments. This issue actually relate to our previous work (Ref. 34, Nature Communications 11, 4415 (2020)) where

demonstration of polaron is well displayed. As the concept causes confusion for the reviewer, we are delighted to discuss the polaron nature again. Firstly, the spin-polarized STM measurements have demonstrated that, the S-terminated surface is not spin-polarized while **the vacancy bound state is spin-polarized**. In addition, the spatial distribution of bound states is directly correlated to the underlying kagome lattice. These results demonstrate that the magnetic moment of the bound state is not result from surface state projected at the S surface but from the localization of the Co *d* electrons by the vacancy potential of the missing S²⁻ ion, which hybridize with the S *p*-electrons. Secondly, the observed anomalous Zeeman response, the energy of bound state linearly shift to higher energy side independent of the direction of magnetic field, implicates that a dominant diamagnetic orbital current contribution. Based on above two conclusions, the polaron nature featured by **the local spin and orbital current** trapped by the lattice distortion around the vacancies due to the spin-orbit coupling. Normally, the spin-trapped polaron is named by magnetic polaron. To emphasize its orbital magnetic moment, we dub the magnetic bound state as a localized spin-orbit polaron (SOP).

As the STM technique is surface sensitive, we are unable to get the experimental evidence that the relaxations persist below the topmost atomic layer. Theoretically calculated Co-Co distance at the CoSn₃ surface underlying S vacancies (0.253 nm) is slightly smaller than the one underlying perfect S surface (0.258 nm), which indicates that the relaxation persist below the top atomic layer.

Accordingly, we have added the definition of SOP to avoid confusions:

“The bound states around the single vacancy have been demonstrated to emergent from the SOP³⁵, where vacancy-potential localized spin from the Co *d* electrons and diamagnetic orbital current are trapped by the lattice distortion around the vacancies”.

Comment 2. *The picture of how the vacancies interact is completely unclear. Fig. 2a,b show that the electronic properties of the vacancies are essentially unchanged until the separation is 2a (this also rules out polaronic effects, as the lateral extension of the polarons should mediate interactions beyond nearest-neighbor distances), when a dramatic shift of the lowest energy peak occurs. Yet, even in Fig. S7 no clear indication of a bonding and anti-bonding peak is visible (none of the states have a nodal plane between the vacancies), pointing to a more complex interaction than simple chemistry based on orbital overlap. For me, this nullifies the data from fig. 3j based on a simple tight-binding model. Also, it seems critical to address the later magnetic observations in fig. 4.*

Response 2. We thank the reviewer for the comments.

(1) *Interaction between two polarons.* The four STM/STS measurements of merging two spatially-separated single vacancies into a dimer vacancy in Fig. 2a and 2b may

mislead the reviewer reach the conclusion that “*the electronic properties of the vacancies are essentially unchanged until the separation is $2a$* ” and “*this also rules out polaronic effects, as the lateral extension of the polarons should mediate interactions beyond nearest-neighbor distances.*” Actually, the nearest-neighbor distances is $\sqrt{3}a$ instead of $2a$ (Fig. S8 (a)). As shown in the following Fig. S8, the lowest energy peak are essentially unchanged until the separation of $3a$. Therefore, the mediate interactions are beyond nearest-neighbor distances, which is consistent with the polaronic effect.

Fig. S8. Two spatially-separated single vacancies with decreasing spacing. **a**, STM image of perfect S surface showing the nearest-neighbor (purple), second-nearest-neighbor (red), third-nearest-neighbor (yellow) and forth-nearest-neighbor atoms (cyan) around the center S atom. ($V_s = -400$ mV, $I_t = 500$ pA). **b,c** STM images of two spatially-separated vacancy with second-nearest-neighbor and forth-nearest-neighbor distance. **d**, dI/dV spectra for two spatially-separated vacancy with decreasing spacing L , showing that the primary peak shifts when $L < 3a$ ($V_s = -300$ mV, $I_t = 500$ pA, $V_{\text{mod}} = 0.5$ mV). The scale bars in all figures correspond to 1 nm.

(2) We agree that none of bound states have a nodal plane between the vacancies point to a more complex interaction than simple chemistry due to the strong spin-orbit coupling in such magnetic Weyl semimetal. This is consistent with experiment-theory discrepancy in the spatial distribution bound states due to the limitation of standard DFT to explain the defect excitations in our previous work (details please refer to Supplementary Fig. 8 in Ref. 34, *Nature Communications* **11**, 4415 (2020)).

However, in Fig. 3j, the tight-binding model is a simple model, which can only explain the evolution of the primary bound states with the vacancy size and shape. We agree that the anti-bonding interaction may over-claim the complex interactions in such system.

Accordingly, we replace the “antibonding interaction” with “**hybridization**” in the revised manuscript.

Comment 3. *I find all data sets showing strain to be extremely unconvincing. The strain data shown in fig. 2 shows nearly all of the intensity in the vacancy region, where the strain would be ill-defined. Second, in fig. 2e and S11, the strain profile of the measured single vacancies varies widely and often appears to simply follow the background. If these arguments are going to be included in the paper, a detailed and thorough analysis of the single and double vacancies must be presented including the quantitative analysis of the strain characteristics for at least 10 members from each type (more would be better). In this way, it would be clear if the strain is statistically significant from the background. This analysis would also have to confirm that the measurements are not influenced by the magnitude of the chosen applied bias.*

Response 3: We thank the reviewer for the comments. We have carefully checked the strain analysis results and would like response as follows:

(1) For the “*ill-defined strain intensity*” issue, we agree that there should be no strain distributed around the vacancy region. The strain analysis applied in this work, geometric phase analysis method, is a standard method based on Fourier transform and diffraction technique, which are commonly used in the field of STM (Refs. 37 and 38) and TEM (*Nano Lett.* 18, 3746 (2018)). When applying anti-FFT, there will be unavoidable “fake” signals inside the vacancy region, which is the concerned by the reviewer. As commonly applied in the TEM community (please refer to Fig. 2D in *Sci. Adv.* 6: eabc2282 (2020)), we intentionally get rid of the “fake” signal inside the vacancy region and added descriptions to avoid confusions in the revised Fig. 2e and Fig. S15 (previous Fig. S11) as follows.

Fig. 2e. e, Strain analysis of the S vacancies. The Symmetric strain maps $S(\mathbf{r}) \equiv u_{aa}(\mathbf{r}) + u_{bb}(\mathbf{r})$ (middle panel) and antisymmetric strain map $U(\mathbf{r}) \equiv u_{aa}(\mathbf{r}) - u_{bb}(\mathbf{r})$ (right panel) are derived from STM topography (left panel), showing that the lattice distortion around the vacancies. **The vacancy regions area overlaid by black hexagons as there are no strains inside vacancies.**

Fig. S15. Analysis of the lattice distortion around S vacancies. The antisymmetric strain map $U(\mathbf{r}) \equiv u_{aa}(\mathbf{r}) - u_{bb}(\mathbf{r})$ (a) are derived from STM topography (b), showing that the relatively-larger lattice distortion localized at the triangular vacancies. **The vacancy regions are overlaid by black hexagons as there are no strains inside vacancies.**

(2) For the “strain variation in the single vacancy” issue, the impression on that “*the strain profile of the measured single vacancies varies widely and often appears to simply follow the background*” may result from the wide range of color bar. We took the reviewer’s suggestions and done the statistic and quantitative analysis over 15 single vacancies to study the strains (Fig. R3), which demonstrates that the strains around single vacancy are statistically significant from the background. The STM images are obtained by using different sample bias, indicates that the chosen applied bias does not significantly affect the strain analysis.

Fig. R3. Symmetric strain (left) and asymmetric strain (rights) analysis of single vacancies in three different surface regions, showing that the strains are significant from the background. The strain intensity is averaged over the vacancy region accept the vacancy center and then normalized by dividing the averaged background. The applied bias for STM images of three surface regions are -400 mV (region 1#), -360 mV (region 2#) and -300 mV (region 3#) (setpoint current is 500 pA), respectively, indicates that the magnitude of chosen applied bias does not significantly affect the strain analysis.

Comment 4. The authors only mention the possibility of repairing existing vacancies, not the ability to define new ones. Does this mean the scale of the structures is limited to the size of the vacancy islands which are naturally occurring? I think this would be a severe limitation for potentially scaling this system beyond the work shown here.

Response 4: The Reviewer are pointing out the reversibility of the manipulation. We have added experimental evidences to prove that we are able to create new vacancies at top surface (new Fig. 1e,f). Therefore, there should not be a severe limitation.

Revised Fig. 1. e,f STM images showing the topography before (a) and after (b) vacancy creation, demonstrating the remove of S atom at surface. g, Schematic showing the remove of S atom from top S layer to fill the vacancy at the bottom S layer.

Comment 5. Spin-sensitive detection of electronic states is difficult because of its convolution with the electronic states – this is particularly true in determining the stabilisation conditions when measuring magnetic systems. In principle, to compare two magnetic measurements, they measurements have to be taken using a stabilisation condition where it is known that no magnetic contrast is present. As the authors don't discuss the stabilisation strategy at all, the utility of the measurements in fig. 2c is called into question. Also, the yellow curves in the bottom panel are completely indistinguishable.

Response 5: We thank the Reviewer's concerns on the comparison of polarization of bound states between single and dimer vacancy. We make such a claim by directly comparing the integrating the difference between the differential conductance between spin-up and spin-down tips, which is just an indication from the spin-sensitive measurement. We agree that the stabilization strategy should be discussed when comparing the spin-polarized dI/dV curves between single and dimer vacancies. As the energy position for the primary bound states around single and dimer vacancy is not the same, the commonly magnetic contrast method is not applicable.

In the present work, our main conclusion is that the bound states at the dimer vacancy is spin polarized, similar to the case of single vacancy. The detailed spin-sensitive comparison among various vacancies are out of range of this work, which is worthy for studying in the future. Thus, we have deleted the sentence of “~~The polarization of bound states for the dimer vacancy is larger than the one for single vacancy~~”, which will not affect the main conclusion of this work.

Comment 6. How are the peak positions for the bound states determined (e.g. fig. 2d, but also for fig. 4)? It seems quite a challenge to me because the peaks are clearly not simple Gaussians or Lorentzians, in fact some look as though they are more than one peak, and it actually seems that most of the curves in fig. 2d actually have a similar onset around -251 meV. It also looks as though the peak width (or perhaps one of the internal peaks) is changing in the magnetic field. Have the authors tried fitting these peaks to determine (1) if they can be fit with single Gaussian or Lorentzian peaks, (2) if the peak widths are changing, and (3) if there are multiple peaks in these ground states. This level of detail would certainly aid in understanding the “shifts” seen with magnetic field and should be examined in more detail.

Response 6: We thank the Reviewer’s suggestive comment.

(1) For the comment “How are the peak positions for the bound states determined?”.

The peaks in Fig. 2d, Fig. 4a and 4b corresponding to localized bound states can be fitted with simple Gaussian by subtracting the same reasonable background (Fig. S16 (a)). However, the broad peak around Fermi level in Fig. 4d corresponding to states of Co_3Sn narrow terraces cannot be fitted by a single Gaussian (consistent with previous STM work on Co_3Sn surface, see *Science* **365**, 1286 (2019)), which may due to the contribution from both itinerant and localized Co 3d electrons of kagome flat band.

(2) For the comment “it actually seems that most of the curves in fig. 2d actually have a similar onset around -251 meV”, it may be the width-height ratio of the Figure make such impression. We now plot the same raw data with larger width-height ratio using both waterfall and color style which show obvious differences for the onset (Fig. R4).

Fig. R4.a, dI/dV spectra of the bound states at dimer vacancy in a magnetic field perpendicular to the sample surface from -8 T to 8 T, showing an approximately linear shift independent of the magnetic field direction. b, Intensity plot of (a), showing the shift of both peak position and peak onset.

(3) For the comments “It also looks as though the peak width (or perhaps one of the internal peaks) is changing in the magnetic field.”. we have checked the evolution of peak width (FWHM) with the magnetic field based on simple Gaussian fitting in Fig. 2d, which shows no significant variations of (Fig. S16 (b)).

We agree with the Reviewer that such level of detail would certainly aid in understanding the “shifts” seen with magnetic field and should be examined in more detail. Accordingly, we have added a new supplementary Figure as new Fig. S16 to add the detailed analysis of the field dependent data as follows.

Fig. S16. Gaussian fitting of the dI/dV spectra in Fig. 2d and Fig. 4d. a, Fitting of one typical dI/dV spectrum in Fig. 2d, showing that the peak can be fitted by a simple Gaussian peak. b, Evolution of FWHM subtracted from Gaussian fitting with magnetic field, showing no significant variation with the field. c, Fitting of one typical dI/dV spectrum in Fig. 2d, showing that the peak cannot be fitted by a simple Gaussian peak.

Comment 7. The negative magnetic moments in this material are ascribed to the orbital magnetization – can the authors explain why this orbital moment would decrease for a single vacancy? Additionally, why should this moment be so different from substitutional In impurities, which show an effective moment of $-5\mu_B$.

Response 7: Compared with single vacancy, there exist interactions between neighboring vacancies, which result in the increasing orbital magnetization with larger vacancy size. That may explain the orbital moment would decrease for a single vacancy. Although the orbital moment for the S vacancy and In substitution are both originated from the spin-polarized *d*-electrons of Co atoms, the detailed orbital hybridization is different. The orbital moment of single vacancy is due to the localization of the Co *d*-

electrons by the vacancy potential of the missing S^{2-} ion, which hybridize with the S p -electrons. In contrast, for the In substitution, In impurity couples with the Co atoms in the kagome lattice, the large effective moment may result from higher orbital angular momentum of the hybridized In-Co orbital (*Nat. Commun.* **11**, 4415 (2020)).

Comment 8. *Additional technical questions:*

8-1: *What is the success rate of vacancy repair? Does it depend on tip state? What is the tip conditioning prior to manipulation?*

Response 8-1: The success rate of the S vacancy repair is about 20% in experiments. The success rate depends on many conditions such as the pulse position, the pulse voltage and the sharpness of tip. The sharpness of tip is interpreted by the spatial resolution of STM topography. Normally, we get a higher success rate when putting on the sharper tip exactly on the center of vacancy and applying the same tip pulse voltage. Accordingly, we have added these information in the method part.

“Vacancy manipulation

Through hundreds times attempts at surfaces of 3 $Co_3Sn_2S_2$ samples, the success rate of the S vacancy repair and creation in experiments is about 30% and 5%, respectively. The success rate depends on many conditions such as the pulse position, the pulse voltage and the sharpness of tip. The sharpness of tip is interpreted by the spatial resolution of STM topography. Normally, we get a higher success rate when putting on the sharper tip exactly on the center of vacancy and applying the same tip pulse voltage.
”

8-2: *The proposed mechanism of sulfur migration would leave a sulfur vacancy in the subsurface layer, it seems surprising that this would not have an electronic effect on the surface properties, have the authors calculated the LDOS for one such sulfur subsurface vacancy?*

Response 8-2: We thank the reviewer for the comment. We agree that the formation of a new sulfur vacancy in the subsurface layer would have an electronic effect on the surface properties. Our claim that “However, after applying the tip pulse, the dI/dV spectrum obtained at the same position exhibits features identical to those of vacancy-free region, providing additional evidence that the S single vacancy is repaired by the tip pulse (Fig. S4).” may mislead the reviewer to reach the point of “*it seems surprising that this would not have an electronic effect on the surface properties*”. In fact, our point is that the spectral features near Fermi level (from -400 meV to 200 meV) for studying vacancy bound states in this work are nearly identical between as-repaired

vacancies and naturally-formed vacancies. There should be spectral differences beyond such energy.

According to the reviewer’s suggestion, we have calculated the LDOS with one sulfur subsurface vacancy and with no vacancies, respectively (Fig. R5). We found that there are no significant differences at the energy ranges from -500 to 500 meV. Although there exist experiment-theory discrepancy due to the limitation of standard DFT to explain the defect excitations in our previous work (details please refer to Supplementary Fig. 8 in Ref. 34, *Nature Communications* **11**, 4415 (2020)), there results may explain very little electronic effect near Fermi level from subsurface S vacancy on the surface properties.

Fig. R5. Comparison of LDOS at perfect S surface without (left panel) and with (right panel) subsurface vacancies, showing similar features near Fermi level as highlighted by the red dotted box.

Accordingly, to avoid confusion, we have revised the sentences to address the identical electronic features is at the energy ranges near Fermi level instead the whole energy range as follows.

“However, after applying the tip pulse, the dI/dV spectrum obtained at the same position exhibits features identical to those of vacancy-free region **near Fermi level**, providing additional evidence that the S single vacancy is repaired by the tip pulse (Fig. S4).”

8-3: Also, is it possible that sulfur is transferred from the tip to the surface? Can the authors rule this out?

Response 8-3: Yes, we can rule out the possibility. In the experiments, the tip used for vacancy repair is freshly fabricated and calibrated at Au surfaces. After calibration, the tip is immediately transferred into the STM chamber for the STM manipulation at the S surface. Upon approaching into the S surface, the tip apex is ultra clean without any S adatoms. At that case, the vacancy repair/creation are achieved, which definitely rule out that the S atom is transferred from the tip to the surface.

Accordingly, we have added the discussion in the main text:

“...The manipulation is achieved by using a clean tip which is immediately transferred into the STM chamber once calibrated at Au surfaces. Therefore, it suggests that the filling S atom originates from the underlying layers rather than the surface or STM tip.”

8-4: Where are the spectra in fig. 2b taken?

Response 8-4: The position of each curve is marked by the blue circle in Fig. 2a. Accordingly, we have added the information in the Figure caption.

“The spatial positions for collecting dI/dV spectra are marked by the black circles at each STM topographic images.”

8-5: Why is the peak spacing strongly modified in the case of the dimer?

Response 8-5: The peak spacing is possibly due to the energy splitting induced by exchange field. The strongly-modified peak spacing in the dimer case possibly result from the stronger exchange field than the one in single vacancy case.

8-6: What direction is up in fig. 2c?

Response 8-6: The direction is defined corresponding to the sample surface. To make the definition of spin clear, we have added the definition in the text and the schematics in Fig. 2c.

“We define the spin direction as “up” and “down” with respect to cleaved S surfaces.”

Revised Fig. 2c. Inset: schematics showing the magnetization of tip and the vacancy bound state.

8-7: Why do the peaks above (although they are lower in absolute energy) the lowest energy bound state change so strongly with varying island/structure size?

Response 8-7: The concerns are similar to the comment 8-5. The satellite peaks above the primary peak (the lowest energy bound states) with equal energy spacing. These peaks change so strongly possible due to the size/shape dependent exchange field.

8-8: What is the size of the error bars in fig. 3i? How many measurements are taken for each data point?

Response 8-8: The size of error bars in Fig. 3i is smaller than 3 meV. For the small atomic numbers vacancies, we have taken tens of measurements. For the large atomic numbers vacancies, we have taken several measurements as the possibility to obtain large vacancy is relatively low.

Accordingly, we have added the information in the Figure caption.

“The error bars in (i) from multiple measurements on the same-geometry vacancies are smaller than 3 meV.”

8-9: How are the error bars in fig. 4e determined? How many measurements are taken per point here?

Response 8-9: The error bars are determined by both linear fitting. For single vacancy, we have taken several measurements (please refer to the main Figure and supplementary in our previous work). For other vacancy, we have just taken once or twice due to the relatively low possibility to obtain large vacancies and long time for taken the field dependent experiments. The error bar is small for server times measurements of single vacancy. In addition, the magnetic moment of step edge is similar the value reported in the literature. Therefore, these data is enough to clarify the evolution of magnetic moment.

Accordingly, we have added the information in the Figure caption.

“The error bars in (e) are determined by linear fitting as shown in bottom panel in (a), (b) and (d).”

REVIEWER COMMENTS

Reviewer #1 (Remarks to the Author):

The revised manuscript addresses the concerns of the reviewers on technical issues regarding data acquisition and interpretation. In my view those questions have been answered satisfactorily. However, I am still concerned with the main outcome of the work presented in this manuscript. The major work presented and discussed in the text and figures deals with the S vacancy including its creation and size and shape dependent spectral properties in tunneling spectroscopy. Despite the timely importance of the material class used in this study and the appealing and curious type of surface modification using STM, I am still left with a feeling that there is no advancement in understanding the nature of the SOP matter. It is certainly worthwhile to publish the vacancy manipulation scheme but I would not recommend to do so in Nature Communications. Despite the claims made in the manuscript, I do not see this as a new platform for engineering atomic quantum states for spintronics and quantum technologies as the authors write. This is rather very vague. Also the discussions are centered on the vacancy manipulation and there is no real indication for designer quantum material nor „practical applications“ as atomic memory or qubits.

This has mainly to do, as pointed out in my first review, with the understanding of the observations, i.e., the apparent magnetic moment of the highest vacancy state. Only half a page is dedicated to this measurement presenting five points in a graph shown in Fig. 4e. As said previously, the physics and discussion of this behavior would, in my view, warrant publication in Nature Communications but apart from the data nothing further is discussed. Hence, what are the conclusions to be drawn for the shape dependent apparent magnetic moments? It cannot be just a sentence that the magnetic moments extend from localized magnetic moment for a vacancy to flat band negative magnetism for an open layer.

Few more comments.

The indication of the spin up and down situation is made more clear in Fig. 2c. However a complete measurement would include the reversal of the surface moment in addition to the two tip magnetic states, i.e., in total four measurements.

The polaron nature of the state is still not clear to me. In any case, the bulk magnetization curve should enter the Supplementary Materials and should also be extended to 8 T, i.e., the same field range as used for the vacancies. This regards the bulk magnetization in comparison to the bulk Co derived magnetic localized states.

All states shift up in energy in an applied magnetic field, however the slope is given as negative value on page 13 first line, whereas in the figure it is positive. The N=infinity state is actually unoccupied rendering the surface non-magnetic?

What does the curve and shaded area in Fig. 4e suggest? There is no theory behind that, right?

Reviewer #2 (Remarks to the Author):

satisfactory responses and revisions. acceptance in NC recommended.

Reviewer #3 (Remarks to the Author):

The authors have addressed a significant portion of my comments from the first submission. I still have a couple of follow up questions/comments:

1. I still question the nomenclature of a polaron for these isolated vacancies, however, these results are published already and not the topic of this work. What I still find hard to follow is the spacing-dependent dimer data in fig. S8. The primary issue I have with this data is that it doesn't follow any clear monotonous trend, as one would expect for an interacting system (i.e. increasing interactions with decreasing separation). For instance, this is most clearly seen in the strong modification of the 2a spectrum, yet only a minor perturbation in the $\sqrt{3}a$ spectrum.
2. To explain the effective behaviour of the vacancy magnetism the authors state "Compared with single vacancy, there exist interactions between neighboring vacancies, which result in the increasing orbital magnetization with larger vacancy size." What types of interactions are the authors referring to precisely? It remains unclear to me why these types of orbital moments should be additive in such a way.

Point-by-point response to the comments from the reviewers

Response to Reviewer 1[#]

Overall Critical Comment:

“The revised manuscript addresses the concerns of the reviewers on technical issues regarding data acquisition and interpretation. In my view those questions have been answered satisfactorily. However, I am still concerned with the main outcome of the work presented in this manuscript. The major work presented and discussed in the text and figures deals with the S vacancy including its creation and size and shape dependent spectral properties in tunneling spectroscopy. Despite the timely importance of the material class used in this study and the appealing and curious type of surface modification using STM, I am still left with a feeling that there is no advancement in understanding the nature of the SOP matter. It is certainly worthwhile to publish the vacancy manipulation scheme but I would not recommend to do so in Nature Communications. Despite the claims made in the manuscript, I do not see this as a new platform for engineering atomic quantum states for spintronics and quantum technologies as the authors write. This is rather very vague. Also the discussions are centered on the vacancy manipulation and there is no real indication for designer quantum material nor “practical applications” as atomic memory or qubits.”

This has mainly to do, as pointed out in my first review, with the understanding of the observations, i.e., the apparent magnetic moment of the highest vacancy state. Only half a page is dedicated to this measurement presenting five points in a graph shown in Fig. 4e. As said previously, the physics and discussion of this behavior would, in my view, warrant publication in Nature Communications but apart from the data nothing further is discussed. Hence, what are the conclusions to be drawn for the shape dependent apparent magnetic moments? It cannot be just a sentence that the magnetic moments extend from localized magnetic moment for a vacancy to flat band negative magnetism for an open layer.

Response: We appreciate the reviewer's satisfactory of technical concerns raised in the previous review. However, we recognize the continued focus on the main outcomes of our work and appreciate the opportunity to clarify and address the concerns.

1. Structural control of vacancies:

(i) Novelty and Significance:

We politely disagree the Reviewer's point that *“It is certainly worthwhile to publish the vacancy manipulation scheme but I would not recommend to do so in Nature Communications.”* Our work introduces a pioneering approach to vacancy manipulation, transcending conventional methods observed in noble metal surfaces

(please see the Review paper: *Nature Reviews Physics*, **1**, 703-715 (2019)). The unique atom migration across layers of the kagome magnet, as demonstrated in this study, provides a new avenue for constructing artificial nanostructures. We note recent works in *Nature Communications* (**14**, 3690, 2023) and *Nature Nanotechnology* (**18**, 1401, 2023) that further underscore the significance of defect manipulation in condensed matter physics, affirming the relevance of our contribution.

(ii) Engineering Quantum States:

Reviewer's concerns that "*Despite the claims made in the manuscript, I do not see this as a new platform for engineering atomic quantum states for spintronics and quantum technologies as the authors write.*" And "*Also the discussions are centered on the vacancy manipulation and there is no real indication for designer quantum material nor "practical applications" as atomic memory or qubits.*" In the field of STM manipulation, incremental progress in step-by-step atomic manipulation on spin-related systems holds promise for engineering quantum states (*Nature Reviews Physics*, **1**, 703-715 (2019)). Our work lays the foundation for the future engineering of atomic quantum states by showcasing the step-by-step manipulation of vacancies. We have extended our efforts to automatically manipulate hundreds of S adatoms or S vacancies, constructing aligned patterns of one-dimensional (1D) atomic chains and 1D atomic vacancies on the surface of the kagome magnet, as illustrated in Figure R1. These achievements demonstrate the potential for designing quantum materials and functional nanostructures.

Fig. R1. Automatic repeated manipulation of S adatoms or S vacancies to construct aligned patterns of 1D atomic chains (a) and 1D atomic vacancies (b) on the surface of kagome magnet $\text{Co}_3\text{Sn}_2\text{S}_2$.

The statement in the manuscript regarding practical applications, such as atomic memory and quantum qubits, is presented as an outlook. If the reviewer finds the phrase

“designer quantum material and practical applications as atomic memory or qubits” too assertive, we are open to revising it to a more cautious tone, such as “...which **may be** promising for practical applications such as atomic memory and quantum qubit.”

2. Shape-dependent bound states and magnetic moments:

The reviewer seeks more conclusive explanations for the shape-dependent apparent magnetic moments, emphasizing the need for in-depth theoretical discussions beyond a single sentence. We acknowledge the complexity of the theoretical aspects in kagome-based materials and recognize the challenges in providing a clear theoretical explanation. The experimental conclusions drawn in our work, including the size-dependent shift in vacancy bound states' energy and the strengthening of local vacancy bound state moments evolving into a magnetic state correlated with the energy band of kagome layers, represent significant advancements. While a comprehensive theoretical description remains elusive, our experimental findings establish a critical connection that will undoubtedly drive further understanding and theoretical investigations in this field.

We hope these clarifications address the reviewer's concerns and contribute to the overall improvement of the manuscript. Accordingly, we have added the following sentences in the discussion part:

“The intriguing evolution of the magnetic moment with increasing vacancy size poses a challenge for theoretical frameworks supporting quantitative model calculations to understand many-body spin-orbit impurities^{34,35} and negative flat band magnetism⁴³ in kagome magnet.”

Comment 1: *The indication of the spin up and down situation is made more clear in Fig. 2c. However, a complete measurement would include the reversal of the surface moment in addition to the two tip magnetic states, i.e., in total four measurements.*

Response: We appreciate the reviewer's observation and suggestion regarding a complete measurement, including the reversal of the surface moment in addition to the two-tip magnetic states. In our previous work, we have successfully demonstrated the determination of the spin-down majority of the bound state by continuously flipping the spin at the tip side, as illustrated in the following Figure. We believe that this approach provides sufficient information for a comprehensive understanding of the spin state associated with the bound state.

It is important to note that the primary focus of the present work is to report the atomically precise manipulation of magnetic bound states on the surface. While the suggested four measurements could systematically study the magnetization curve of the

bound state, it falls outside the specific scope of this work. We acknowledge the value of such a detailed investigation for future research, and we appreciate the suggestion for additional measurements to further explore the magnetization behavior of the bound states. However, for the current study, we believe that the continuous spin flipping at the tip side sufficiently characterizes the spin states associated with the bound states, aligning with the primary objectives of this research endeavor.

Fig. 2c in our previous work: Spin-flip operation of the STM tip and the reproducible spectra at the S vacancy site. Left curve (curve I) corresponds to the initial spin-down tip polarization, the middle one (curve II) corresponds to spin-up tip polarization induced by a magnetic field, and the right one (curve III) corresponds to flipping the spin of the tip back to the initial spin-down polarization. ($V_s = -400$ mV, $I_t = 500$ pA, $V_{\text{mod}} = 0.5$ mV)

Comment 2: *The polaron nature of the state is still not clear to me. In any case, the bulk magnetization curve should enter the Supplementary Materials and should also be extended to 8 T, i.e., the same field range as used for the vacancies. This regards the bulk magnetization in comparison to the bulk Co derived magnetic localized states.*

Response: We appreciate the reviewer's suggestion and agree that including the bulk magnetization curve in the Supplementary Materials is a valuable addition to enhance the comprehensive understanding of our findings. In response to this suggestion, we have incorporated the bulk magnetization curve in Fig. S19.

It's important to note that the maximum field range available in our instrument for magnetization measurements is 5 T. Consequently, the presented magnetization curve covers the range from -5 T to 5 T, aligning with the field range used for the investigation

of vacancies in the dI/dV spectra. We believe this range provides sufficient data for meaningful comparisons and analysis.

We trust that the inclusion of the bulk magnetization curve in the Supplementary Materials will contribute to a more comprehensive interpretation of our results, addressing the reviewer's concern regarding the comparison between the field-dependent dI/dV spectra and the bulk Co-derived magnetic localized states.

New Fig. S19. Comparison with magnetic field dI/dV spectra obtained at S vacancies of $\text{Co}_3\text{Sn}_2\text{S}_2$ surface (a) and magnetization of bulk $\text{Co}_3\text{Sn}_2\text{S}_2$ crystal (b). The coercive field is approximately 4500 Oe, and the saturation magnetization is 10.38 emu/g. The magnetic moment corresponding to a cobalt atom is about $0.3 \mu_B$.

Comment 3: All states shift up in energy in an applied magnetic field, however the slope is given as negative value on page 13 first line, whereas in the figure it is positive. The $N=\infty$ state is actually unoccupied rendering the surface non-magnetic?

Response: The reviewer correctly points out the discrepancy between the negative slope mentioned in the text on page 13 and the positive slope observed in Figure 4. We appreciate the clarification opportunity and would like to provide a more accurate explanation.

The negative value of the slope in the text is expressed in terms of the magnetic moment (μ) and is intended to emphasize the anomalous Zeeman effect. This is crucial as a normal Zeeman effect would result in a positive magnetic moment. In contrast, our measurements reveal a peak shifting towards positive energies, indicating an effective negative magnetic moment of the state. To avoid confusion, we have revised the sentences as follows:

“ $\mu(N=\infty) = -0.19 \text{ meV/T} = -3.28\mu_B$ (the negative value is aiming to highlight the moment is negative in anomalous Zeeman effect and differentiate it from the positive moment in Zeeman effect)”

The label of Y-axis in Figure 4e is changed to “Absolute value of magnetic moment $|\mu|$ (μ_B)”

Regarding the concern about the unoccupied $N=\infty$ state, we present the magnetic field response of the highest-energy peak for $N=\infty$ state to facilitate comparison with the local moment for $N=\text{finite}$ values. It's essential to note that showcasing this state does not imply that the Co_3Sn surface is non-magnetic. In reality, there are several spin-polarized states at both occupied and unoccupied energy levels, consistent with theoretical calculations indicating spin polarization on the Co_3Sn surface (Nat. Phys. **14**, 1125–1131, 2018).

Comment 4: *What does the curve and shaded area in Fig. 4e suggest? There is no theory behind that, right?*

Response: The curve and shaded area in Figure 4e are provided as a visual guide to the trend in the evolution of the moment with increasing atomic number N . It is important to note that, as mentioned in response to the Overall Critical Comment, there is currently no specific theory explaining this behavior. The visual representation is intended to aid in the qualitative understanding of the observed trends.

We acknowledge the absence of a comprehensive theoretical explanation at this stage. However, we believe that the experimental conclusions themselves represent a significant advancement in this field. The complexities in the theoretical aspect highlight the challenges in the current state of the field, and we anticipate that our experimental findings will contribute to and stimulate further understanding and theoretical developments in the future. We appreciate the reviewer's understanding of these challenges and hope that the visual representation serves its intended purpose of illustrating the observed trends in the absence of a detailed theoretical framework.

Accordingly, to avoid the misunderstanding of the curve and shaded line, we added the sentence in the caption of Figure 4(e):

“e, ...vacancies to flat band negative magnetism from Co_3Sn kagome layer. The colored shade curve is just visual guide to the trend in the evolution of the moment with increasing atomic number N .”

Response to Reviewer 2[#]

Satisfactory responses and revisions. acceptance in NC recommended.

Response: We thank the Reviewer for positive remarks and recommendation.

Response to Reviewer 3[#]

The authors have addressed a significant portion of my comments from the first submission. I still have a couple of follow up questions/comments:

Response: We appreciate that the Reviewer is satisfied with a significant portion of his comments. We have also addressed all the concerns from the Reviewer as follows.

Comment 1: *I still question the nomenclature of a polaron for these isolated vacancies, however, these results are published already and not the topic of this work. What I still find hard to follow is the spacing-dependent dimer data in fig. S8. The primary issue I have with this data is that it doesn't follow any clear monotonous trend, as one would expect for an interacting system (i.e. increasing interactions with decreasing separation). For instance, this is most clearly seen in the strong modification of the $2a$ spectrum, yet only a minor perturbation in the $\sqrt{3}a$ spectrum.*

Response: Fig. S8 is shown to address the Reviewer's previous question on the nearest two single vacancies. In an ideal one-dimensional system, the Reviewer correctly pointed out the monotonous trend for the two vacancies getting closer. However, in the S surface of $\text{Co}_3\text{Sn}_2\text{S}_2$ with six-fold symmetry, the system is more complicate. Except for the distance, the effect of orientation for the two separated vacancies may also matters, that is why we choose the two separated single vacancies with the similar orientation ($d=2a, 3a, 4a\dots$) in Fig. 2a. The atomic configuration of two separated single vacancies for the $\sqrt{3}a$ case is distinct from the case of $2a$, which may explain the difference between these two cases.

Comment 2: *To explain the effective behaviour of the vacancy magnetism the authors state "Compared with single vacancy, there exist interactions between neighboring vacancies, which result in the increasing orbital magnetization with larger vacancy size." What types of interactions are the authors referring to precisely? It remains unclear to me why these types of orbital moments should be additive in such a way.*

Response: We thank the Reviewer for pointing out this concern regarding the statement on interactions between neighboring vacancies. Upon reflection, we recognize that the term "interaction" may potentially mislead readers to think of long-range magnetic orders. To provide a more accurate representation, we would like to replace the term "interaction" with "coupling."

In the context of our study, when two vacancies come into proximity, the wavefunctions of the local states at each vacancy overlap, leading to a coupling between the two polarons. Additionally, when two single vacancies combine to form a dimer vacancy, there are changes in the atomic structures of the vacancies. This evolution is expected,

resulting in the observation of both the isolated polarons and the fusion of moments in the dimer vacancy.

To avoid any potential misunderstanding, we have updated the wording from "interaction" to "coupling" in our response. It's essential to note that the term "ferromagnetic" is used here more as a descriptive analogy rather than a precise characterization of the coupling type. We are focused on presenting the experimental discovery of new phenomena, and currently, there is no existing theoretical work or simple model that can fully explain our experimental findings. We believe that this experimental work will inspire and guide future theoretical investigations into the observed phenomena.

Revisions:

“The geometry-dependent bound states suggest the strong ~~interactions~~ couplings between adjacent single vacancies”

“...indicating the existences of vacancy-vacancy ~~interactions~~ couplings.”

“The ~~interactions~~ couplings among the vacancies in $\text{Co}_3\text{Sn}_2\text{S}_2$ could...”

REVIEWERS' COMMENTS

Reviewer #1 (Remarks to the Author):

The authors have further revised the manuscript in view of the additional reviewers comments. I agree with all the modifications. The arguments given in the rebuttal give reasons for the presented vacancy manipulation scheme such that I would leave the impact of this work regarding the understanding of the materials properties to the scientific community. Therefore, I can recommend the publication of this manuscript in Nature Communications.

Reviewer #3 (Remarks to the Author):

The authors have satisfactorily responded to all of my comments.